# On the Pitfalls of Heteroscedastic Uncertainty Estimation with Probabilistic Neural Networks

**Maximilian Seitzer**[1]**, Arash Tavakoli**[1]**, Dimitrije Antić**[2]**, Georg Martius**[1]

[1] Max Planck Institute for Intelligent Systems, Tübingen, Germany
[2] University of Tübingen, Tübingen, Germany
`maximilian.seitzer@tuebingen.mpg.de`

## Abstract

Capturing aleatoric uncertainty is a critical part of many machine learning systems. In deep learning, a common approach to this end is to train a neural network to estimate the parameters of a heteroscedastic Gaussian distribution by maximizing the logarithm of the likelihood function under the observed data. In this work, we examine this approach and identify potential hazards associated with the use of log-likelihood in conjunction with gradient-based optimizers. First, we present a synthetic example illustrating how this approach can lead to very poor but stable parameter estimates. Second, we identify the culprit to be the log-likelihood loss, along with certain conditions that exacerbate the issue. Third, we present an alternative formulation, termed $\beta-$NLL, in which each data point's contribution to the loss is weighted by the $\beta$-exponentiated variance estimate. We show that using an appropriate $\beta$ largely mitigates the issue in our illustrative example. Fourth, we evaluate this approach on a range of domains and tasks and show that it achieves considerable improvements and performs more robustly concerning hyperparameters, both in predictive RMSE and log-likelihood criteria.

## 1 Introduction

Endowing models with the ability to capture *uncertainty* is of crucial importance in machine learning. Uncertainty can be categorized into two main types: *epistemic* uncertainty and *aleatoric* uncertainty (Kiureghian & Ditlevsen, 2009). Epistemic uncertainty accounts for subjective uncertainty in the model, one that is reducible given sufficient data. By contrast, aleatoric uncertainty captures the stochasticity inherent in the observations and can itself be subdivided into *homoscedastic* and *heteroscedastic* uncertainty. Homoscedastic uncertainty corresponds to noise that is constant across the input space, whereas heteroscedastic uncertainty corresponds to noise that varies with the input.

There are well-established benefits for modeling each type of uncertainty. For instance, capturing epistemic uncertainty enables effective budgeted data collection in active learning (Gal et al., 2017), allows for efficient exploration in reinforcement learning (Osband et al., 2016), and is indispensable in cost-sensitive decision making (Amodei et al., 2016). On the other hand, quantifying aleatoric uncertainty enables learning of dynamics models of stochastic processes (e.g. for model-based or offline reinforcement learning) (Chua et al., 2018; Yu et al., 2020), improves performance in semantic segmentation, depth regression and object detection (Kendall & Gal, 2017; Harakeh & Waslander, 2021), and allows for risk-sensitive decision making (Dabney et al., 2018; Vlastelica et al., 2021).

We examine a common approach for quantifying aleatoric uncertainty in neural network regression. By assuming that the regression targets follow a particular distribution, we can use a neural network to predict the parameters of that distribution, typically the input-dependent mean and variance when assuming a heteroscedastic Gaussian distribution. Then, the parameters of the network can be learned using maximum likelihood estimation (MLE), i.e. by minimizing the negative log-likelihood (NLL) criterion using stochastic gradient descent. This simple procedure, which is the de-facto standard (Nix & Weigend, 1994; Lakshminarayanan et al., 2017; Kendall & Gal, 2017; Chua et al., 2018), is known to be subject to overconfident variance estimates. Whereas strategies have been proposed to alleviate this specific issue (Detlefsen et al., 2019; Stirn & Knowles, 2020), we argue that an equally important

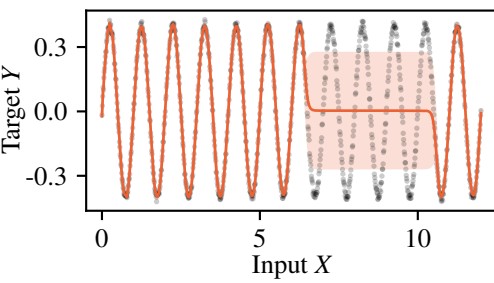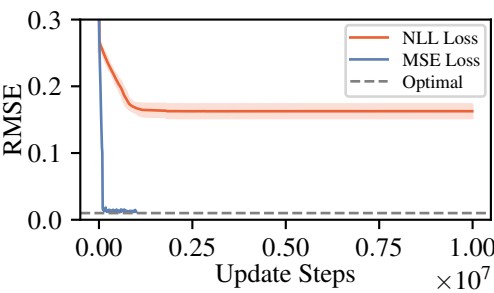

Figure 1: Training a probabilistic neural network to fit a simple sinusoidal fails. Left: Learned predictions (orange line) after $10^7$ updates, with the shaded region showing the predicted standard deviation. The target function is given by $y(x) = 0.4\sin(2\pi x) + \xi$, where $\xi$ is Gaussian noise with a standard deviation of $0.01$. Right: Root mean squared error (RMSE) over training, mean and standard deviation over 10 random seeds. For comparison, we plot the training curve when using the mean squared error as the training objective – achieving an optimal mean fit (dashed line) in $10^5$ updates. This behavior is stable across different optimizers, hyperparameters, and architectures (see Sec. B.2).

issue is that this procedure can additionally lead to *subpar mean fits*. In this work, we analyze and propose a simple modification to mitigate this issue.

**Summary of contributions** We demonstrate a pitfall of optimizing the NLL loss for neural network regression, one that hinders the training of accurate mean predictors (see Fig. 1 for an illustrative example). The primary culprit is the *high dependence of the gradients on the predictive variance*. While such dependence is generally known to be responsible for instabilities in joint optimization of mean and variance estimators (Takahashi et al., 2018; Stirn & Knowles, 2020), we identify a fresh perspective on how this dependence can further be problematic. Namely, we hypothesize that the issue arises due to the NLL loss scaling down the gradient of poorly-predicted data points relative to the well-predicted ones, leading to effectively undersampling the poorly-predicted data points.

We then introduce an *alternative loss formulation*, termed $\beta-$NLL, that counteracts this by weighting the contribution of each data point to the overall loss by its $\beta$-exponentiated variance estimate, *where $\beta$ controls the extent of dependency of gradients on predictive variance*. This formulation subsumes the standard NLL loss for $\beta = 0$ and allows to lessen the dependency of gradients on the variance estimates for $0 < \beta \leq 1$. Interestingly, using $\beta = 1$ completely removes such dependency for training the mean estimator, yielding the standard mean squared error (MSE) loss – but with the additional capacity of uncertainty estimation. Finally, we empirically show that our modified loss formulation largely mitigates the issue of poor fits, achieving considerable improvements on a range of domains and tasks while exhibiting more robustness to hyperparameter configurations.

## 2 PRELIMINARIES

Let $X, Y$ be two random variables describing the input and target, following the joint distribution $P(X, Y)$. We assume that $Y$ is conditionally independent given $X$ and that it follows some probability distribution $P(Y \mid X)$. In the following, we use the common assumption that $Y$ is normally distributed given $X$; i.e. $P(Y \mid X) = \mathcal{N}(\mu(X), \sigma^2(X))$, where $\mu \colon \mathbb{R}^M \mapsto \mathbb{R}$ and $\sigma^2 \colon \mathbb{R}^M \mapsto \mathbb{R}^+$ are respectively the true input-dependent mean and variance functions.[1] Equivalently, we can write $Y = \mu(X) + \epsilon(X)$, with $\epsilon(X) \sim \mathcal{N}(0, \sigma^2(X))$; i.e. $Y$ is generated from $X$ by $\mu(X)$ plus a zero-mean Gaussian noise with variance $\sigma^2(X)$. This input-dependent variance quantifies the heteroscedastic uncertainty or input-dependent aleatoric uncertainty.

To learn estimates $\hat{\mu}(X), \hat{\sigma}^2(X)$ of the true mean and variance functions, it is common to use a neural network $f_\theta$ parameterized by $\theta$. Here, $\hat{\mu}(X)$ and $\hat{\sigma}^2(X)$ can be outputs of the final layer (Nix & Weigend, 1994) or use two completely separate networks (Detlefsen et al., 2019). The variance output is hereby constrained to the positive region using a suitable activation function, e.g. softplus.

---

[1]For notational convenience, we focus on univariate regression but point out that the work extends to the multivariate case as well.

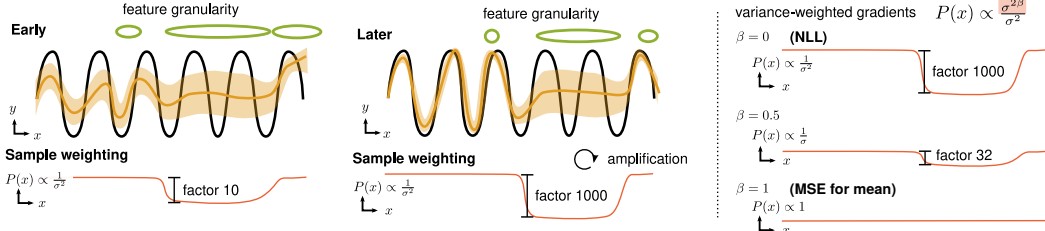

Figure 2: Illustration of the pitfall when training with NLL (negative log-likelihood) versus our solution. An initial inhomogeneous feature space granularity (see Sec. 3.1) results early on in different fitting quality. The implicit weighting of the squared error in NLL can be seen as biased data-sampling with $p(x) \propto \frac{1}{\sigma^2(x)}$ (see Eq. 6). Badly fit parts are increasingly ignored during training. On the right, the effect of our solution (Eq. 7) on the relative importance of data points is shown.

The optimal parameters $\theta^*_{\text{NLL}}$ can then be found using maximum likelihood estimation (MLE) by minimizing the negative log-likelihood (NLL) criterion $\mathcal{L}_{\text{NLL}}$ under the distribution $P(X, Y)$:

$$\theta^*_{\text{NLL}} = \arg\min_{\theta} \mathcal{L}_{\text{NLL}}(\theta) = \arg\min_{\theta} \mathop{\mathbb{E}}_{X,Y}\left[\frac{1}{2}\log\hat{\sigma}^2(X) + \frac{(Y - \hat{\mu}(X))^2}{2\hat{\sigma}^2(X)} + \text{const}\right]. \quad (1)$$

In contrast, standard regression minimizes the mean squared error (MSE) $\mathcal{L}_{\text{MSE}}$:

$$\theta^*_{\text{MSE}} = \arg\min_{\theta} \mathcal{L}_{\text{MSE}}(\theta) = \arg\min_{\theta} \mathop{\mathbb{E}}_{X,Y}\left[\frac{(Y - \hat{\mu}(X))^2}{2}\right]. \quad (2)$$

In practice, Eq. 1 and Eq. 2 are optimized using stochastic gradient descent (SGD) with mini-batches of samples drawn from $P(X, Y)$. The gradients of $\mathcal{L}_{\text{NLL}}$ w.r.t. (with respect to) $\hat{\mu}(X), \hat{\sigma}^2(X)$ are given by

$$\nabla_{\hat{\mu}}\mathcal{L}_{\text{NLL}}(\theta) = \mathop{\mathbb{E}}_{X,Y}\left[\frac{\hat{\mu}(X) - Y}{\hat{\sigma}^2(X)}\right], \quad \nabla_{\hat{\sigma}^2}\mathcal{L}_{\text{NLL}}(\theta) = \mathop{\mathbb{E}}_{X,Y}\left[\frac{\hat{\sigma}^2(X) - (Y - \hat{\mu}(X))^2}{2(\hat{\sigma}^2(X))^2}\right]. \quad (3, 4)$$

## 3 ANALYSIS

We now return to the example of trying to fit a sinusoidal function from Sec. 1. Recall from Fig. 1 that using the Gaussian NLL as the objective resulted in a suboptimal fit. In contrast, using MSE as the objective the model converged to the optimal mean fit in a reasonable time. We now analyze the reasons behind this surprising result.

From Eq. 3, we see that the true mean $\mu(X)$ is the minimizer of the NLL loss. It thus becomes clear that a) the solution found in Fig. 1 is not the optimal one, and b) the NLL objective should, in principle, drive $\hat{\mu}(X)$ to the optimal solution $\mu(X)$. So, why does the model not converge to the optimal solution? We identify two main culprits for this behavior of the Gaussian NLL objective:

1. Initial flatness of the feature space can create an undercomplex but locally stable mean fit. This fit results from local symmetries and requires a form of symmetry breaking to escape.
2. The NLL loss scales the gradient of badly-predicted points down relative to well-predicted points, effectively undersampling those points. This effect worsens as training progresses.

These culprits and their effect on training are illustrated in Fig. 2 (left). If the network cannot fit a certain region yet because its feature space (spanned by the last hidden layer) is too coarse, it would then perceive a high effective data variance. This leads to down-weighting the data from such regions, fueling a vicious cycle of self-amplifying the increasingly imbalanced weighting. In the following, we analyze these effects and their reasons in more detail.

### 3.1 SYMMETRY AND FEATURE NON-LINEARITY

It is instructive to see how the model evolves during training as shown in Fig. 3. The network first learns essentially the best linear fit while adapting the variance to match the residuals. The situation

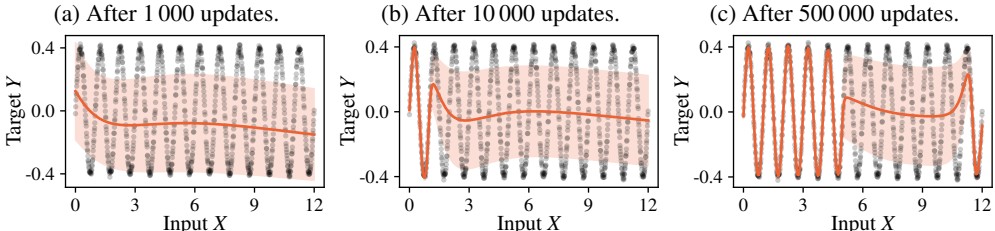

Figure 3: Model fit using the NLL loss at different stages of training shown in orange with $\pm\sigma$ uncertainty band. Black dots mark training data. Fitting the function begins from the left and is visibly slow.

is locally stable. That is, due to the symmetries of errors below and above the mean fit, there is no incentive to change the situation. Symmetry breaking is required for further progress. One form of symmetry breaking comes with the inherent stochasticity of mini-batch sampling in SGD, or the natural asymmetries contained in the dataset due to, e.g., outliers. Moreover, we hypothesize that the local non-linearity of the feature space plays an important role in creating the necessary non-linear fit.

Let us consider the non-linearity of the feature space. This quantity is not easy to capture. To approximate it for a dataset $\mathcal{D}$, we compute how much the Jacobian $J_f$ of the features $f(x)$ w.r.t. the input varies in an $L_2$-ball with radius $r$ around a point $x$, denoted as the Jacobian variance:[2]

$$V(x) = \frac{1}{|\mathcal{B}_x|} \sum_{x' \in \mathcal{B}_x} \left( J_f(x') - \frac{1}{|\mathcal{B}_x|} \sum_{x'' \in \mathcal{B}_x} J_f(x'') \right)^2, \quad \mathcal{B}_x = \{x' \in \mathcal{D}: \|x - x'\|_2 \leq r\}. \quad (5)$$

Figure 4 visualizes the Jacobian variance over the input space as a function of the training progress. Although initially relatively flat, it becomes more granular in parts of the input space, the parts which are later well fit. The region with low Jacobian variance remains stuck in this configuration (see Fig. 1). This provides evidence that the non-linearity of the feature space is important for success or failure of learning on this dataset. However, why does gradient descent not break out of this situation?

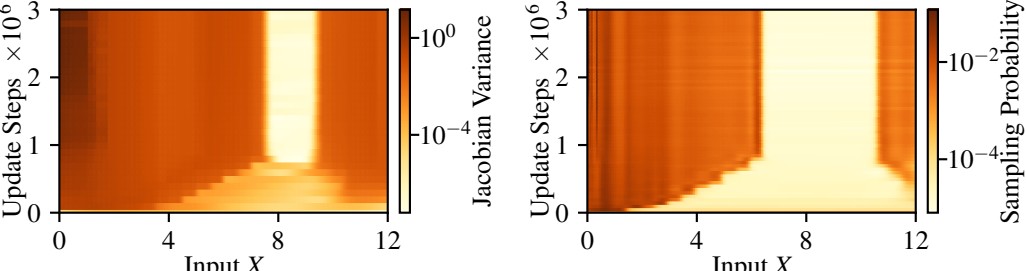

Figure 4: Jacobian variance over training time, using the mean of matrix $V(x)$ (see Eq. 5).

Figure 5: Probability of sampling a data point at input $x$ over training time.

## 3.2 Inverse-Variance Weighting Effectively Undersamples

The answer lies in an imbalanced weighting of data points across the input space. Recall that the gradient $\nabla_{\hat{\mu}} \mathcal{L}_{\text{NLL}}$ of the NLL w.r.t. the mean scales the error $\hat{\mu}(X) - Y$ by $\frac{1}{\hat{\sigma}^2(X)}$ (Eq. 3). As symmetry is broken and the true function starts to be fit locally, the variance quickly shrinks in these areas to match the reduced MSE. If the variance is well-calibrated, the gradient becomes $\frac{\hat{\mu}(X)-Y}{\hat{\sigma}^2(X)} \approx \frac{\hat{\mu}(X)-Y}{(\hat{\mu}(X)-Y)^2} = \frac{1}{\hat{\mu}(X)-Y}$. Data points with already low error will get their contribution in the batch gradient scaled up relatively to high error data points – "rich get richer" self-amplification. Thus, NLL acts contrary to MSE which focuses on high-error samples. If the true variance $\sigma^2$ on the well-fit regions is much smaller than the errors on the badly-fit regions, or there are much more well-fit than badly-fit points, then *learning progress is completely hindered* on the badly-fit regions.

---

[2]This is a form of approximative second-order derivative computed numerically, which also gives non-zero results for networks with relu activation (in contrast to, for example, the Hessian).

Another way to view this is to interpret the different weighting of points as *changing the training distribution $P(X, Y)$ to a modified distribution $\tilde{P}(X, Y)$* in which points with high error have a lower probability of getting sampled. This can be shown by defining $\tilde{P}(X, Y) = Z^{-1} \frac{P(X,Y)}{\sigma^2(X)}$, where $Z = \int \frac{P(x,y)}{\sigma^2(x)} \mathrm{d}x \mathrm{d}y$ is a normalizing constant, and recognizing that the gradient of the NLL is proportional to the gradient of the MSE loss in Eq. 2 under the modified data distribution $\tilde{P}(X, Y)$:

$$\nabla_{\hat{\mu}} \mathcal{L}_{\mathrm{NLL}}(\theta) = Z \cdot \mathbb{E}_{X,Y \sim \tilde{P}(X,Y)}[\hat{\mu}(X) - Y] \propto \nabla_{\hat{\mu}} \mathbb{E}_{X,Y \sim \tilde{P}(X,Y)} \left[ \frac{(Y - \hat{\mu}(X))^2}{2} \right]. \tag{6}$$

In Fig. 5, we plot $\tilde{P}(X, Y)$ over training time for our sinusoidal example. It can be seen that the virtual probability of sampling a point from the high-error region drops over time until it is highly unlikely to sample points from this region ($10^{-5}$ as opposed to $10^{-3}$ for uniform sampling). We show that this behavior also carries over to a real-world dataset in Sec. B.3.

Sometimes, "inverse-variance weighting" is seen as a feature of the Gaussian NLL (Kendall & Gal, 2017) which introduces a self-regularizing property by allowing the network to "ignore" outlier points with high error. This can be desirable if the predicted variance corresponds to data-inherent unpredictability (noise), but it is undesirable if it causes premature convergence and ignorance of hard-to-fit regions, as shown above. In our method, we enable control over the extent of self-regularization.

## 4 METHOD

In this section, we develop a solution method to mitigate these issues with NLL training. Our approach, which we term $\beta-$NLL, allows choosing an arbitrary loss-interpolation between NLL and MSE while keeping calibrated uncertainty estimates.

### 4.1 VARIANCE-WEIGHTING THE GRADIENTS OF THE NLL

The problem we want to address is the premature convergence of NLL training to highly suboptimal mean fits. In Sec. 3, we identified the relative down-weighting of badly-fit data points in the NLL loss together with its self-amplifying characteristic as the main culprit. Effectively, NLL weights the mean-squared-error per data point with $\frac{1}{\sigma^2}$, which can be interpreted as sampling data points with $P(x) \propto \frac{1}{\sigma^2}$. Consequently, we propose modifying this distribution by introducing a parameter $\beta$ allowing to *interpolate* between NLL's and a completely uniform data point importance. The resulting sampling distribution is given by $P(x) \propto \frac{\sigma^{2\beta}}{\sigma^2}$ and illustrated in Fig. 2 (right).

How could this weighting be achieved? We simply introduce the variance-weighting term $\sigma^{2\beta}$ to the $\mathcal{L}_{\mathrm{NLL}}$ loss such that it acts as a factor on the gradient. We denote the resulting loss as $\beta-$NLL:

$$\mathcal{L}_{\beta-\mathrm{NLL}} := \mathbb{E}_{X,Y} \left[ \lfloor \hat{\sigma}^{2\beta}(X) \rfloor \left( \frac{1}{2} \log \hat{\sigma}^2(X) + \frac{(Y - \hat{\mu}(X))^2}{2\hat{\sigma}^2(X)} + \mathrm{const} \right) \right], \tag{7}$$

where $\lfloor \cdot \rfloor$ denotes the *stop gradient* operation. By stopping the gradient, the variance-weighting term acts as an adaptive, input-dependent learning rate. In this way, the gradients of $\mathcal{L}_{\beta-\mathrm{NLL}}$ are:

$$\nabla_{\hat{\mu}} \mathcal{L}_{\beta-\mathrm{NLL}}(\theta) = \mathbb{E}_{X,Y} \left[ \frac{\hat{\mu}(X) - Y}{\hat{\sigma}^{2-2\beta}(X)} \right], \quad \nabla_{\hat{\sigma}^2} \mathcal{L}_{\beta-\mathrm{NLL}}(\theta) = \mathbb{E}_{X,Y} \left[ \frac{\hat{\sigma}^2(X) - (Y - \hat{\mu}(X))^2}{2\hat{\sigma}^{4-2\beta}(X)} \right]. \tag{8, 9}$$

Naturally, for $\beta = 0$, we recover the original NLL loss. For $\beta = 1$ the gradient w.r.t. $\mu$ in Eq. 8 is equivalent to the one of MSE. However, for the variance, the gradient in Eq. 9 is a new quantity with $2\sigma^2$ in the denominator. For values $0 < \beta < 1$, we get different loss interpolations. Particularly interesting is the case of $\beta = 0.5$, where the data points are weighted with $\frac{1}{\sigma}$ (inverse standard deviation instead of inverse variance). In our experiments (Sec. 5), we find that $\beta = 0.5$ generally achieves the best trade-off between accuracy and log-likelihood. A Pytorch implementation of the loss function is provided in Sec. D.5.

Note that the new loss $\mathcal{L}_{\beta-\mathrm{NLL}}$ is not meant for performance evaluation, rather it is designed to result in meaningful gradients. Due to the weighting term, the loss value does not reflect the model's quality. The model performance during training should be monitored with the original negative log-likelihood objective and, optionally, with RMSE for testing the quality of the mean fit.

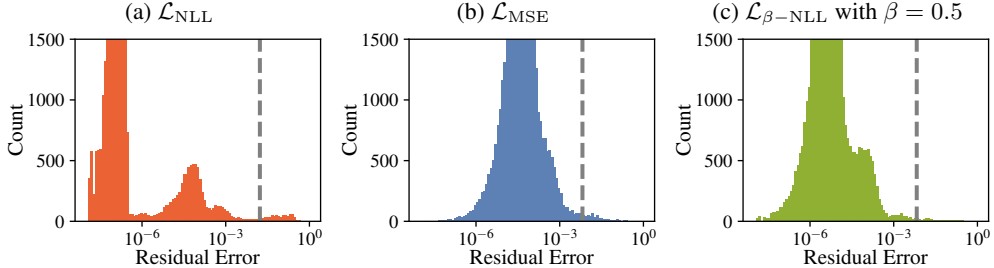

Figure 6: Distribution of residual prediction errors depending on the loss function for the ObjectSlide dataset (see Sec. 5.2). Dashed lines show predictive RMSE. (a) The NLL loss ($\mathcal{L}_{\mathrm{NLL}}$) yields multimodal residuals. There is a long tail of difficult data points that are ignored, while easy ones are fit to high accuracy. (b) The MSE loss ($\mathcal{L}_{\mathrm{MSE}}$) results in a log-normal residual distribution. (c) Our $\beta-$NLL loss ($\mathcal{L}_{\beta-\mathrm{NLL}}$) yields highly accurate fits on easy data points without ignoring difficult ones.

## 4.2 ALLOCATION OF FUNCTION APPROXIMATOR CAPACITY

Even though $\mathcal{L}_{\mathrm{MSE}}$, $\mathcal{L}_{\mathrm{NLL}}$, and $\mathcal{L}_{\beta-\mathrm{NLL}}$ all have the same optima w.r.t. the mean (and also the variance in the case of $\mathcal{L}_{\mathrm{NLL}}$ and $\mathcal{L}_{\beta-\mathrm{NLL}}$), optimizing them leads to very different solutions. In particular, because these losses weight data points differently, *they assign the capacity of the function approximator differently*. Whereas the MSE loss gives the same weighting to all data points, the NLL loss gives high weight to data points with low predicted variance and low weight to those with high variance. $\beta-$NLL interpolates between the two. The behavior of the NLL loss is appropriate if these variances are caused by true aleatoric uncertainty in the data. However, due to the use of function approximation, there is also the case where data points cannot be well predicted (maybe only transiently). This would result in high predicted variance, although the ground truth is corrupted by little noise. The different loss functions thus vary in how they handle these *difficult data points*.

An example of how the differences between the losses manifest in practice is illustrated in Fig. 6. Here we show the distribution of the residuals for a dynamics prediction dataset containing *easy* and *hard* to model areas. The NLL loss essentially ignores a fraction of the data by predicting high uncertainty. By analyzing those data points, we found that they were actually *the most important data points* to model correctly (because they captured non-trivial interactions in the physical world).

How important are the data points with high uncertainty? Are they outliers (i.e. do they stem from truly noisy regions), to which we would be willing to allocate less of the function approximator's capacity? Or are they just difficult samples that are important to fit correctly? The answer is task-dependent and, as such, there is no one-loss-fits-all solution. Rather, the modeler should choose which behavior is desired. Our $\beta-$NLL loss makes this choice available through the $\beta$ parameter.

## 5 EXPERIMENTS

In our experiments, we ask the following questions and draw the following conclusions:

Sec. 5.1: *Does $\beta-$NLL fix the pitfall with NLL's convergence?*
**Yes**, $\beta-$NLL converges to good mean and uncertainty estimates across a range of $\beta$ values.

Sec. 5.2: *Does $\beta-$NLL improve over NLL in practical settings? How sensitive to hyperparameters is $\beta-$NLL?* We investigate a diverse set of real-world domains: regression on the UCI datasets, dynamics model learning, generative modeling on MNIST and Fashion-MNIST, and depth-map prediction from natural images.
**Yes**, $\beta-$NLL generally performs better than NLL and is considerably easier to tune.

Sec. 5.3: *How does $\beta-$NLL compare to other loss functions for distributional regression?* We compare with a range of approaches: learning to match the moments of a Gaussian (termed "moment matching" (MM); see Sec. A), using a Student's t-distribution instead of a Gaussian (Detlefsen et al., 2019), or putting different priors on the variance and using variational inference (xVAMP, xVAMP*, VBEM, VBEM*) (Stirn & Knowles, 2020).
**It depends.** Different losses make different trade-offs, which we discuss in Sec. 5.3.

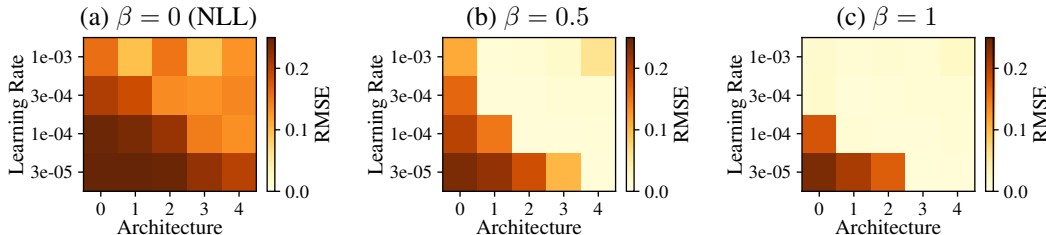

Figure 7: Convergence properties analyzed on the sinusoidal regression problem. RMSE after 200 000 epochs, averaged over 3 independent trials, is displayed by color codes (lighter is better) as a function of learning rate and model architecture (see Sec. D.1). The original NLL ($\beta = 0$) does not obtain good RMSE fits for most hyperparameter settings. Figure S3 shows results for the NLL metric.

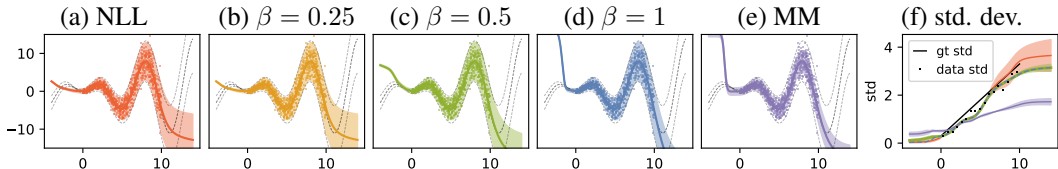

Figure 8: Fits for the heteroscedastic sine example from Detlefsen et al. (2019) (a-e). Dotted lines show the ground truth mean and $\pm 2\sigma$, respectively. (f) The predicted standard deviations (with shaded std. over 10 independent trials) with the same color code. Note that $\beta = 0.5$ and $\beta = 1$ graphs lie on top of one another. Inside the training regime, all $\beta-$NLL variants (a-d) yield well-calibrated uncertainty estimates. Moment matching (e) significantly underestimates the variance everywhere.

We refer the reader to Sec. C and Sec. D for a description of datasets and training settings. We evaluate the quality of predictions quantitatively in terms of the root mean squared error (RMSE) and the negative log-likelihood (NLL).

## 5.1 SYNTHETIC DATASETS

**Sinusoidal without heteroscedastic noise** We first perform an extended investigation of our illustrative example from Fig. 1 – a sine curve with a small additive noise: $y = 0.4\sin(2\pi x) + \xi$, with $\xi$ being Gaussian noise with standard deviation $\sigma = 0.01$. One would expect that a network with sufficient capacity can easily learn to fit this function. Figure 7 inspects this over a range of architectures and learning rates.

We find that for the standard NLL loss ($\beta = 0$), the networks do not converge to a reasonable mean fit. There is a trend that larger networks and learning rates show better results, but when comparing this to $\beta-$NLL with $\beta = 0.5$ we see that the networks are indeed able to fit the function without any issues. As expected, the same holds for the mean squared error loss (MSE) and $\beta-$NLL with $\beta = 1$. The quality of the fit w.r.t. NLL is shown in Fig. S3.

**Sinusoidal with heteroscedastic noise** We sanity-check that $\beta-$NLL is still delivering good uncertainty estimates on the illustrative example from Detlefsen et al. (2019) – a sine curve with increasing amplitude and noise: $y = x\sin(x) + x\xi_1 + \xi_2$, with $\xi_1$ and $\xi_2$ being Gaussian noise with standard deviation $\sigma = 0.3$. Figure 8 (a-e) displays the predictions of the best models (w.r.t. NLL validation loss) and (f) compares the predicted uncertainties over 10 independent trials. Fitting the mean is achieved with all losses. On the training range, $\beta-$NLL with $\beta > 0$ learns virtually the same uncertainties as the NLL loss.

## 5.2 REAL-WORLD DATASETS

**UCI Regression Datasets** As a standard real-world benchmark in predictive uncertainty estimation, we consider the UCI datasets (Hernández-Lobato & Adams, 2015). Table 1 gives an overview comparing different loss variants. We refer to Sec. B.4 for the full results on all 12 datasets. The results are encouraging: $\beta-$NLL achieves predictive log-likelihoods on par with or better than the NLL loss while clearly improving the predictive accuracy on most datasets.

Table 1: Results for UCI Regression Datasets. We report predictive log-likelihood and RMSE ($\pm$ standard deviation). *Ties* denotes the number of datasets (out of 12) for which the method cannot be statistically distinguished from the best method (see Sec. B.4). We compare with Student-t (Detlefsen et al., 2019) and xVAMP/VBEM (Stirn & Knowles, 2020). Section B.4 lists the full results.

| Loss | $\beta$ | LL ↑ | | | | | RMSE ↓ | | | | |
| | | Ties | concrete | energy | naval | yacht | Ties | concrete | energy | naval | yacht |
|---|---|---|---|---|---|---|---|---|---|---|---|
| $\mathcal{L}_{\beta-\text{NLL}}$ | 0 | 3 | -3.25 ± 0.31 | -3.22 ± 1.41 | 12.46 ± 1.18 | -2.86 ± 5.18 | 5 | 6.08 ± 0.65 | 2.25 ± 0.34 | 0.0021 ± 0.0006 | 1.22 ± 0.47 |
| $\mathcal{L}_{\beta-\text{NLL}}$ | 0.25 | 4 | -3.31 ± 0.51 | -2.82 ± 0.82 | 13.78 ± 0.33 | -1.97 ± 1.14 | 6 | 5.79 ± 0.74 | 1.81 ± 0.30 | 0.0012 ± 0.0004 | 1.73 ± 1.00 |
| $\mathcal{L}_{\beta-\text{NLL}}$ | 0.5 | 5 | -3.29 ± 0.36 | -2.41 ± 0.72 | 13.99 ± 0.40 | -2.47 ± 1.68 | 7 | 5.61 ± 0.65 | 1.12 ± 0.25 | 0.0006 ± 0.0002 | 2.35 ± 1.44 |
| $\mathcal{L}_{\beta-\text{NLL}}$ | 0.75 | 5 | -3.27 ± 0.34 | -2.80 ± 0.59 | 13.63 ± 0.62 | -1.87 ± 0.55 | 8 | 5.67 ± 0.73 | 1.31 ± 0.45 | 0.0004 ± 0.0001 | 1.97 ± 1.03 |
| $\mathcal{L}_{\beta-\text{NLL}}$ | 1.0 | 2 | -3.23 ± 0.33 | -3.37 ± 0.58 | 13.59 ± 0.30 | -2.27 ± 1.07 | 9 | 5.55 ± 0.77 | 1.54 ± 0.54 | 0.0004 ± 0.0000 | 2.08 ± 1.13 |
| $\mathcal{L}_{\text{MM}}$ | | 0 | -3.49 ± 0.38 | -4.26 ± 0.50 | 12.73 ± 0.64 | -11.2 ± 31.0 | 5 | 6.28 ± 0.82 | 2.19 ± 0.28 | 0.0005 ± 0.0001 | 3.02 ± 1.38 |
| $\mathcal{L}_{\text{MSE}}$ | | — | — | — | — | — | 12 | 4.96 ± 0.64 | 0.92 ± 0.11 | 0.0004 ± 0.0001 | 0.78 ± 0.25 |
| Student-t | | 10 | -3.07 ± 0.14 | -2.46 ± 0.34 | 12.47 ± 0.48 | -1.23 ± 0.55 | 5 | 5.82 ± 0.59 | 2.26 ± 0.34 | 0.0026 ± 0.0009 | 1.34 ± 0.63 |
| xVAMP | | 6 | -3.06 ± 0.15 | -2.47 ± 0.32 | 12.44 ± 0.60 | -0.99 ± 0.33 | 8 | 5.44 ± 0.64 | 1.87 ± 0.32 | 0.0023 ± 0.0004 | 0.99 ± 0.43 |
| xVAMP* | | 7 | -3.03 ± 0.13 | -2.41 ± 0.32 | 12.80 ± 0.55 | -1.04 ± 0.47 | 8 | 5.35 ± 0.73 | 2.00 ± 0.26 | 0.0020 ± 0.0006 | 1.13 ± 0.66 |
| VBEM | | 2 | -3.14 ± 0.07 | -4.29 ± 0.16 | 8.05 ± 0.13 | -2.65 ± 0.10 | 9 | 5.21 ± 0.58 | 1.29 ± 0.33 | 0.0009 ± 0.0004 | 1.66 ± 0.84 |
| VBEM* | | 8 | -2.99 ± 0.13 | -1.91 ± 0.21 | 13.10 ± 0.47 | -0.98 ± 0.24 | 9 | 5.17 ± 0.59 | 1.08 ± 0.17 | 0.0015 ± 0.0005 | 0.65 ± 0.20 |

(a) ObjectSlide       (b) Fetch-PickAndPlace

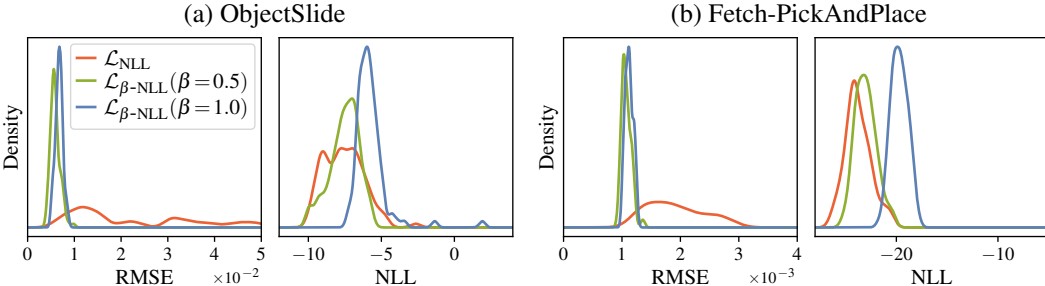

Figure 9: Sensitivity analysis of loss functions to hyperparameters on the dynamics model learning tasks: ObjectSlide (a) and Fetch-PickAndPlace (b). The distributions over validation RMSE and NLL are shown as a function of hyperparameters, based on a grid search over different model configurations (see Sec. D.2). While the NLL loss is highly sensitive when evaluating RMSE, the $\beta-$NLL loss shows much less sensitivity and yields good results regardless of the exact configuration.

**Dynamics models** As a major application of uncertainty estimation lies in model-based reinforcement learning (RL), we test the different loss functions on two dynamics predictions tasks of varying difficulty, ObjectSlide, and Fetch-PickAndPlace. In both tasks, the goal is to predict how an object will move from the current state and the agent's action. Whereas ObjectSlide (Seitzer et al., 2021) is a simple 1D-environment, Fetch-PickAndPlace (Plappert et al., 2018) is a complex 3D robotic-manipulation environment. The models are trained on trajectories collected by RL agents.

For both datasets, we perform a grid search over different hyperparameter configurations (see Sec. D.2) for a sensitivity analysis to hyperparameters settings, presented in Fig. 9. It reveals that NLL is vulnerable to the choice of hyperparameters, whereas $\beta-$NLL achieves good results over a wide range of configurations. The best performing configurations for each loss are then evaluated on a hold-out test set (Table 2). One can see that the NLL loss results in poor predictive performance and also exhibits quite a high variance across random seeds. Our method yields high accuracy and log-likelihood fits for a range of $\beta$ values, with $\beta = 0.5$ generally achieving the best trade-off.

**Generative modeling and depth-map prediction** For generative modeling, we train variational autoencoders (Kingma & Welling, 2014) with probabilistic decoders on MNIST and Fashion-MNIST. For the task of depth regression, we modify a state-of-the-art method (AdaBins; Bhat et al. (2021)) and test it on the NYUv2 dataset (Silberman et al., 2012) with our loss (Fig. S6). Table 3 presents selected results for both tasks, yielding similar trends as before. We refer to Sec. B.5 and Sec. B.6 for more details, including qualitative results.

### 5.3 COMPARISON TO OTHER LOSS FUNCTIONS

The previous sections have demonstrated that our $\beta-$NLL loss has clear advantages over the NLL loss. However, the comparison to other loss functions requires a more nuanced discussion. First,

Table 2: Test results for dynamics models, using best configurations found in a grid search. The reported standard deviations are over 5 random seeds. We compare with Student-t (Detlefsen et al., 2019) and xVAMP/VBEM (Stirn & Knowles, 2020).

| Loss | $\beta$ | 1D-Slide | | Fetch-PickAndPlace | |
|---|---|---|---|---|---|
| | | RMSE $\downarrow$ | LL $\uparrow$ | RMSE $\downarrow$ | LL $\uparrow$ |
| $\mathcal{L}_{\beta-\mathrm{NLL}}$ | 0 | $0.0192 \pm 0.006$ | $7.97 \pm 3.62$ | $0.00163 \pm 0.00008$ | $18.72 \pm 7.32$ |
| $\mathcal{L}_{\beta-\mathrm{NLL}}$ | 0.25 | $0.0107 \pm 0.004$ | $9.03 \pm 0.47$ | $0.00102 \pm 0.00004$ | $24.43 \pm 1.64$ |
| $\mathcal{L}_{\beta-\mathrm{NLL}}$ | 0.5 | $0.0064 \pm 0.002$ | $9.28 \pm 0.75$ | $0.00096 \pm 0.00002$ | $24.68 \pm 0.08$ |
| $\mathcal{L}_{\beta-\mathrm{NLL}}$ | 0.75 | $0.0087 \pm 0.003$ | $6.61 \pm 1.83$ | $0.00098 \pm 0.00001$ | $22.77 \pm 0.17$ |
| $\mathcal{L}_{\beta-\mathrm{NLL}}$ | 1.0 | $0.0074 \pm 0.001$ | $6.58 \pm 0.29$ | $0.00102 \pm 0.00001$ | $21.32 \pm 0.07$ |
| $\mathcal{L}_{\mathrm{MM}}$ | | $0.0078 \pm 0.001$ | diverges | $0.00104 \pm 0.00003$ | $19.33 \pm 1.31$ |
| $\mathcal{L}_{\mathrm{MSE}}$ | | $0.0068 \pm 0.001$ | — | $0.00103 \pm 0.00000$ | — |
| Student-t | | $0.0155 \pm 0.006$ | $11.30 \pm 0.03$ | $0.00117 \pm 0.00001$ | $30.44 \pm 0.08$ |
| xVAMP | | $0.0118 \pm 0.002$ | $10.58 \pm 0.19$ | $0.00128 \pm 0.00005$ | $29.02 \pm 0.12$ |
| xVAMP* | | $0.0199 \pm 0.006$ | $10.89 \pm 0.10$ | $0.00128 \pm 0.00001$ | $29.19 \pm 0.08$ |
| VBEM | | $0.0039 \pm 0.000$ | $3.79 \pm 0.00$ | $0.00104 \pm 0.00003$ | $17.39 \pm 0.29$ |
| VBEM* | | $0.0280 \pm 0.011$ | $10.13 \pm 0.49$ | $0.00118 \pm 0.00003$ | $28.62 \pm 0.15$ |

Table 3: Selected results for generative modeling and depth-map prediction. Left: Training variational autoencoders on MNIST and Fashion-MNIST. Right: Depth-map prediction on NYUv2. Full results can be found in Table S3 and Table S4.

| Loss | $\beta$ | MNIST | | Fashion-MNIST | | Loss | $\beta$ | NYUv2 | |
|---|---|---|---|---|---|---|---|---|---|
| | | RMSE $\downarrow$ | LL $\uparrow$ | RMSE $\downarrow$ | LL $\uparrow$ | | | RMSE $\downarrow$ | LL $\uparrow$ |
| $\mathcal{L}_{\beta-\mathrm{NLL}}$ | 0 | $0.237 \pm 0.002$ | $2116 \pm 55$ | $0.170 \pm 0.001$ | $1940 \pm 104$ | $\mathcal{L}_{\beta-\mathrm{NLL}}$ | 0 | 0.3854 | -4.52 |
| $\mathcal{L}_{\beta-\mathrm{NLL}}$ | 0.5 | $0.151 \pm 0.003$ | $2220 \pm 25$ | $0.125 \pm 0.003$ | $1639 \pm 52$ | $\mathcal{L}_{\beta-\mathrm{NLL}}$ | 0.5 | 0.3789 | -7.50 |
| $\mathcal{L}_{\beta-\mathrm{NLL}}$ | 1.0 | $0.152 \pm 0.001$ | $1706 \pm 30$ | $0.138 \pm 0.002$ | $1142 \pm 26$ | $\mathcal{L}_{\beta-\mathrm{NLL}}$ | 1.0 | 0.3845 | -5.10 |
| Student-t | | $0.273 \pm 0.002$ | $4291 \pm 103$ | $0.182 \pm 0.001$ | $2857 \pm 9$ | $\mathcal{L}_{\mathrm{MSE}}$ | | 0.3776 | — |
| xVAMP* | | $0.225 \pm 0.001$ | $3062 \pm 215$ | $0.160 \pm 0.002$ | $2150 \pm 131$ | $\mathcal{L}_1$ | | 0.3850 | — |
| VBEM* | | $0.176 \pm 0.008$ | $3213 \pm 238$ | $0.150 \pm 0.003$ | $2244 \pm 78$ | SI Loss | | 0.419 | — |

we also test an alternative loss function based on matching the moments of the Gaussian ($\mathcal{L}_{\mathrm{MM}}$; see Sec. A). While this loss results in high accuracy, it is unstable to train and exhibits poor likelihoods.

Second, we test several loss functions based on a Student's t-distribution, including xVAMP and VBEM (Stirn & Knowles, 2020). These approaches generally achieve better likelihoods than the $\beta-\mathrm{NLL}$; we conjecture this is because their ability to maintain uncertainty about the variance results in a better fit (in terms of KL divergence) when the variance is wrongly estimated. In terms of predictive accuracy, $\beta-\mathrm{NLL}$ outperforms the Student's t-based approaches. Exceptions are some of the UCI datasets with limited data (e.g. "concrete" and "yacht") where xVAMP and VBEM are on par or better than $\beta-\mathrm{NLL}$. This is likely because these methods can mitigate overfitting by placing a prior on the variance. However, xVAMP and VBEM are also non-trivial to implement and computationally heavy: both need MC samples to evaluate the prior; for xVAMP, training time roughly doubles as evaluating the prior also requires a second forward pass through the network. In contrast, $\beta-\mathrm{NLL}$ is simple to implement (see Sec. D.5) and introduces no additional computational costs.

## 6 CONCLUSION

We highlight a problem frequently occurring when optimizing probabilistic neural networks using the common NLL loss: training gets stuck in suboptimal function fits. With our analysis, we reveal the underlying reason: initially badly-fit regions receive increasingly less weight in the loss which results in premature convergence. We propose a simple solution by introducing a family of loss functions called $\beta-\mathrm{NLL}$. Effectively, the gradient of the original NLL loss is scaled by the $\beta$-exponentiated per-sample variance. This allows for a meaningful interpolation between the NLL and MSE loss functions while providing well-behaved uncertainty estimates. The hyperparameter $\beta$ gives practitioners the choice to control the self-regularization strength of NLL: how important should high-noise regions or difficult-to-predict data points be in the fitting process. In most cases, $\beta = 0.5$ will be a good starting point. We think the problem discussed in this paper is primarily why practitioners using the Gaussian distribution in regression or generative modeling tasks often opt for a constant or homoscedastic (global) variance, as opposed to the more general heteroscedastic (data-dependent) variance. We hope that our simple solution contributes to changing this situation by improving the usability and performance of modeling data uncertainty with deep neural networks.

## ACKNOWLEDGMENTS

The authors thank the International Max Planck Research School for Intelligent Systems (IMPRS-IS) for supporting Maximilian Seitzer. Georg Martius is a member of the Machine Learning Cluster of Excellence, EXC number 2064/1 – Project number 390727645. We acknowledge the financial support from the German Federal Ministry of Education and Research (BMBF) through the Tübingen AI Center (FKZ: 01IS18039B).

## REPRODUCIBILITY STATEMENT

All settings are described in detail in Sec. C and Sec. D. We make full code and data available under https://github.com/martius-lab/beta-nll.

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

# APPENDIX

## A The Moment Matching Loss

We also investigated an alternative loss function designed to counter the imbalanced weighting of data points by the NLL loss, which we call "moment matching" (MM). It uses standard squared error losses to estimate the *moments of the target distribution*, i.e. directly estimating the sufficient statistics of the target distribution. In our experiments, we found that this loss function generally fixes the problems with premature convergence when using $\mathcal{L}_{\mathrm{NLL}}$, but it also leads to underestimation of variances and exhibits considerable training instabilities.

To fully describe a Gaussian distribution, only the first two moments $\mu, \sigma^2$ need to be estimated. So, why not directly define losses based on the moment estimators? Starting from the conditional mean of the targets, $\mathbb{E}[Y \mid X]$, we can define the squared deviation as a loss and see that the MSE is an upper bound:

$$\left(\mathbb{E}[Y \mid X] - \hat{\mu}(X)\right)^2 = \mathbb{E}[(Y - \hat{\mu}(X)) \mid X]^2 \leq \mathbb{E}\left[(Y - \hat{\mu}(X))^2 \mid X\right] := \mathcal{L}_{\mathrm{MM}}^{\hat{\mu}}. \quad \text{(S1)}$$

Notice that $\mathcal{L}_{\mathrm{MM}}^{\hat{\mu}}$ is the standard MSE loss. Thus, learning a mean fit $\hat{\mu}(x)$ can follow the standard (non-distributional) regression procedures. Interestingly, due to the moment matching viewpoint, we can analogously define a loss for fitting the variance (or the second central moment). Variance is defined as $\mathbb{E}\left[(Y - \mu(X))^2 \mid X\right]$. As such, by analogy, we can define the following loss: $\mathcal{L}_{\mathrm{MM}}^{\hat{\sigma}^2} := \mathbb{E}\left[\left((Y - \hat{\mu}(X))^2 - \hat{\sigma}^2(X)\right)^2 \mid X\right]$. To use the same physical unit as $\mathcal{L}_{\mathrm{MM}}^{\hat{\mu}}$, we reformulate it in terms of the standard deviation as:

$$\mathcal{L}_{\mathrm{MM}}^{\hat{\sigma}} := \mathbb{E}\left[\left(\sqrt{(Y - \hat{\mu}(X))^2} - \hat{\sigma}(X)\right)^2 \mid X\right]. \quad \text{(S2)}$$

Thus, the moment matching loss can be expressed simply by the sum of the two losses: $\mathcal{L}_{\mathrm{MM}} = \mathcal{L}_{\mathrm{MM}}^{\hat{\mu}} + \mathcal{L}_{\mathrm{MM}}^{\hat{\sigma}}$. We found that using $\mathcal{L}_{\mathrm{MM}}^{\hat{\sigma}}$ instead of $\mathcal{L}_{\mathrm{MM}}^{\hat{\sigma}^2}$ makes it easier to balance the losses.

Interestingly, our $\beta-$NLL subsumes the moment matching loss if we allow for different values of $\beta$ to be used for mean and variance estimation. In particular, using $\beta-$NLL with $\beta = 1$ for the mean and $\beta = 2$ for the variance results in the same gradients as using $\frac{1}{2}\mathcal{L}_{\mathrm{MM}}^{\hat{\mu}} + \frac{1}{4}\mathcal{L}_{\mathrm{MM}}^{\hat{\sigma}^2}$ as the loss. Note that we did not investigate this connection further, and also did not perform any experiments with different values of $\beta$ for mean and variance.

## B Additional Results

### B.1 Further Consequences of Inverse-Variance Weighting

In Sec. 3.2, we interpreted the inverse-variance weighting of the NLL as training under a different data distribution $\tilde{P}(X, Y)$ in which points with high error have a low probability of being sampled. A consequence of this is that the distribution $\tilde{P}(X, Y)$ continuously shifts while the model is improving its fit. For datasets where the noise level is low on parts of the input space and the underlying function can be modeled to high accuracy, an interesting phenomenon can be observed: even though the training curve for RMSE looks like it indicates convergence (i.e. it flattens out), the model can actually still be adapting with the changing training distribution. It is just that the changes happen on data points that already have such low prediction error relative to the average error that further improvements are virtually invisible in the average error. The actual training progress can be revealed using a histogram of the prediction errors on a log-scale, in the manner of Fig. 6. Eventually, the training distribution stabilizes when for all data points, either the prediction error reaches the noise level at that point, or the variance can not decrease further at that point because it reaches a manually set lower bound. If no lower bound on the variance is set, the distribution may never stabilize, leading to training instabilities when the variance reaches close to zero.

## B.2 Synthetic Dataset

In Fig. S1, we replicate the experiment from Fig. 1 (training the NLL loss on a sinusoidal for $10^7$ updates) on several more random seeds. In order to test the dependence of our reported issue on the optimizer, we repeat the experiment from Fig. 1 but use different optimizers than Adam with $\beta_1 = 0.9, \beta_2 = 0.999$. The results are shown in Fig. S2. We find that none of the configurations reaches below an RMSE of $0.1$ (the optimal value corresponds to an RMSE of $0.01$), indicating that the issue is occurring stably across optimization settings. In Fig. S3, we provide the results for the NLL metric on the sinusoidal dataset, complementing the results of Fig. 7 (see discussion in Sec. 5.1).

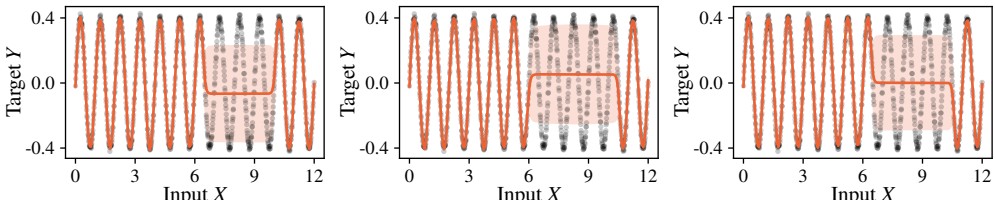

Figure S1: Repeating the experiment from Fig. 1, i.e. training with NLL loss for $10^7$ update steps. The observed behavior is stable across different independent trials.

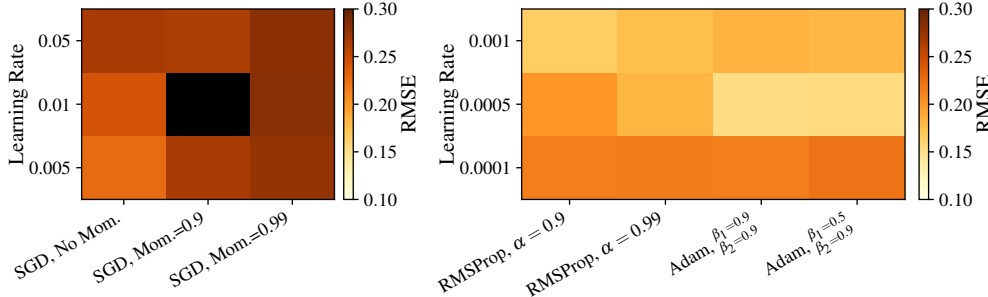

Figure S2: Using different optimizers to train on the sinusoidal from Fig. 1 with the NLL loss. The color code indicates the mean RMSE over 3 independent trials per optimizer setting. Black indicates that all trials diverged. Training was done for $2 \cdot 10^6$ update steps and used architecture 2 from Table S5. The observed behavior is stable across optimization settings.

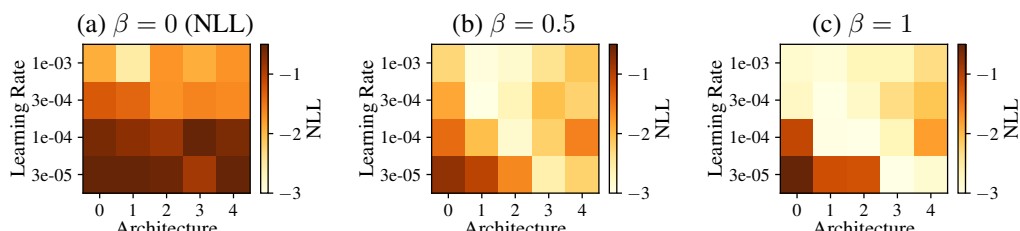

Figure S3: Convergence properties analyzed on the sinusoidal toy regression problem. Same as Fig. 7 but for the negative log-likelihood (NLL) criterion. Due to the bad mean fit, the original NLL loss ($\beta = 0$) is also bad for most hyperparameter settings. With $\beta > 0.5$ good fits are obtained for many settings. Also for $\beta = 1$, which corresponds to MSE for fitting the mean, good uncertainty predictions are obtained with our $\beta-$NLL as testified by the low NLL scores.

## B.3 Analysis of Sampling Probabilities on Fetch-PickAndPlace

In Fig. S4, we show how the analysis from Sec. 3.2 transfers to a real world dataset, namely Fetch-PickAndPlace. The figure shows how the distribution of effective sampling probability evolves during training (over a fixed set of training points) and compares that against a proxy "oracle": the distribution of effective sampling probabilities when using the squared residuals from a model trained

with the MSE loss. The mismatch between the two distributions demonstrates that optimizing the NLL loss drastically undersamples in comparison to the reference, effectively never sampling some data points. This further corroborates our analysis in Sec. 3.2.

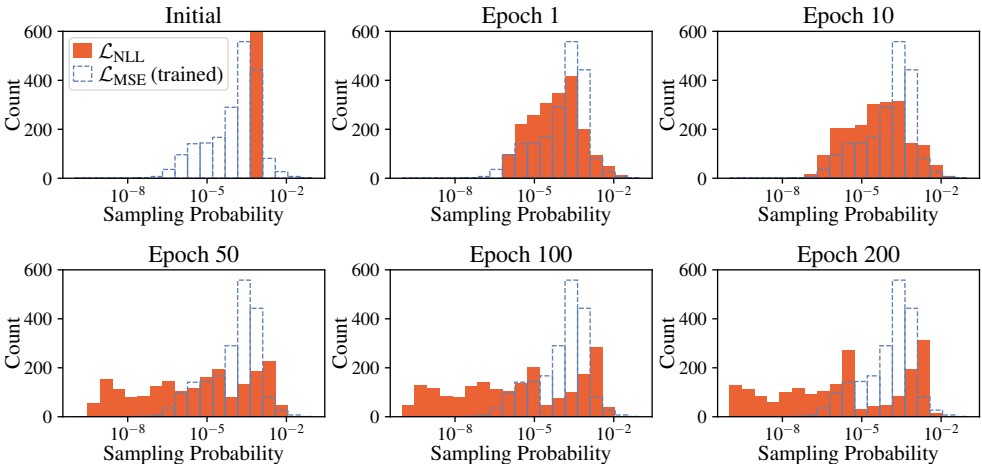

Figure S4: Undersampling behavior of $\mathcal{L}_{\mathrm{NLL}}$ on Fetch-PickAndPlace. The plot shows how the distribution of effective sampling probability evolves over training time, taken over $2\,000$ fixed training points sampled at the initial epoch. The dashed blue histogram shows the distribution of effective sampling probabilities when using the squared residuals from a model trained with MSE loss $\mathcal{L}_{\mathrm{MSE}}$. This gives a reference distribution that $\mathcal{L}_{\mathrm{NLL}}$ should roughly match, taking into account the relative hardness of prediction on different samples. The $\mathcal{L}_{\mathrm{NLL}}$ drastically undersamples compared to the reference (note the log-scale), effectively never sampling some points.

### B.4 UCI DATASETS

In this section, we include results for predictive log-likelihood (Table S1) and RMSE (Table S2) for all UCI datasets we evaluated on.

We find that baselines based on the Student's t-distribution (xVAMP and VBEM (Stirn & Knowles, 2020)) tend to have better predictive log-likelihood than $\beta-$NLL, although there is also a dataset where $\beta-$NLL performs better ("naval") or where there is no statistically significant improvement ("housing", "kin8m", "wine-red", "wine-white"). For RMSE, $\mathcal{L}_{\mathrm{MSE}}$ unsurprisingly performs best. Our $\beta-$NLL is often on par with $\mathcal{L}_{\mathrm{MSE}}$ and on par or better than the other baselines, except for "yacht". At the same time, our method is very simple to implement (see Sec. D.5) and computationally lightweight compared to xVAMP and VBEM, which require the costly evaluation of a prior and Monte-Carlo sampling at each training step.

### B.5 GENERATIVE MODELING WITH VARIATIONAL AUTOENCODERS

We test different loss functions on the task of generative modeling using variational autoencoders (VAEs) (Kingma & Welling, 2014). To this end, we parameterize the decoder distribution $p(x \mid z)$ with $\mathcal{N}\big(\mu(z), \sigma^2(z)\big)$, where the mean $\mu(z)$ and variance $\sigma^2(z)$ are outputs of a neural network. We train the VAE by maximizing the ELBO $\mathbb{E}_{q(z|x)}[\log p(x \mid z)] - D_{\mathrm{KL}}(q(z \mid x) \,\|\, p(z))$, plugging in different loss functions for $\log p(x \mid z)$. Following Stirn & Knowles (2020), we evaluate the log-posterior predictive likelihood $\log \mathbb{E}_{q(z|x)}[p(x \mid z)]$. We approximate the expectation using a finite mixture of 20 Monte-Carlo samples from $q(z \mid x)$. To compute the RMSE, we take the mean of that mixture. We compare $\beta-$NLL against $\mathcal{L}_{\mathrm{MM}}$, $\mathcal{L}_{\mathrm{NLL}}$ with a fixed variance of 1, Student-t (Takahashi et al., 2018; Detlefsen et al., 2019), xVAMP and VBEM (Stirn & Knowles, 2020).

We evaluate on MNIST and FashionMNIST. Table S3 presents quantitative results. VBEM achieves the best reconstruction error at the expense of poor log-likelihood. Vice versa, Student-t, xVAMP, xVAMP*, and VBEM* achieve strong log-likelihoods but worse reconstruction errors. $\beta-$NLL with $\beta > 0$ provides a good compromise between log-likelihood and reconstruction error. Figure S5

Table S1: Results for UCI Regression Datasets. Predictive log-likelihood (higher is better) and standard deviation, together with dataset size, input and output dimensions. Best mean value in bold. Results that are not statistically distinguishable from the best result are marked with †.

| | carbon (10721, 5, 3) | concrete (1030, 8, 1) | energy (768, 8, 2) | housing (506, 13, 1) |
|---|---|---|---|---|
| $\mathcal{L}_{\beta-\text{NLL}}(\beta=0)$ | $11.36 \pm 2.11$ | $-3.25 \pm 0.31$ | $-3.22 \pm 1.41$ | $-2.86 \pm 0.50$ |
| $\mathcal{L}_{\beta-\text{NLL}}(\beta=0.25)$ | $10.91 \pm 2.42$ | $-3.31 \pm 0.51$ | $-2.82 \pm 0.82$ | $-2.75 \pm 0.42$ |
| $\mathcal{L}_{\beta-\text{NLL}}(\beta=0.5)$ | $10.22 \pm 4.00$ | $-3.29 \pm 0.36$ | $-2.41 \pm 0.72$ | $-2.64 \pm 0.36^{\dagger}$ |
| $\mathcal{L}_{\beta-\text{NLL}}(\beta=0.75)$ | $10.82 \pm 1.37$ | $-3.27 \pm 0.34$ | $-2.80 \pm 0.59$ | $-2.72 \pm 0.42^{\dagger}$ |
| $\mathcal{L}_{\beta-\text{NLL}}(\beta=1.0)$ | $3.31 \pm 20.88$ | $-3.23 \pm 0.33$ | $-3.37 \pm 0.58$ | $-2.85 \pm 0.87$ |
| $\mathcal{L}_{\text{MM}}$ | $4.72 \pm 5.80$ | $-3.49 \pm 0.38$ | $-4.26 \pm 0.50$ | $-3.42 \pm 1.01$ |
| $\mathcal{L}_{\text{MSE}}$ | — | — | — | — |
| Student-t | $\mathbf{15.59 \pm 0.43}$ | $-3.07 \pm 0.14^{\dagger}$ | $-2.46 \pm 0.34$ | $-2.47 \pm 0.24^{\dagger}$ |
| xVAMP | $13.28 \pm 0.19$ | $-3.06 \pm 0.15^{\dagger}$ | $-2.47 \pm 0.32$ | $-2.43 \pm 0.21^{\dagger}$ |
| xVAMP* | $13.17 \pm 0.26$ | $-3.03 \pm 0.13^{\dagger}$ | $-2.41 \pm 0.32$ | $-2.45 \pm 0.22^{\dagger}$ |
| VBEM | $5.68 \pm 0.70$ | $-3.14 \pm 0.07$ | $-4.29 \pm 0.16$ | $-2.56 \pm 0.15$ |
| VBEM* | $13.23 \pm 0.36$ | $\mathbf{-2.99 \pm 0.13}$ | $\mathbf{-1.91 \pm 0.21}$ | $\mathbf{-2.42 \pm 0.22}$ |

| | kin8m (8192, 8, 1) | naval (11934, 16, 2) | power (9568, 4, 1) | protein (45730, 9, 1) |
|---|---|---|---|---|
| $\mathcal{L}_{\beta-\text{NLL}}(\beta=0)$ | $1.140 \pm 0.039^{\dagger}$ | $12.46 \pm 1.18$ | $-2.807 \pm 0.057$ | $-2.80 \pm 0.05$ |
| $\mathcal{L}_{\beta-\text{NLL}}(\beta=0.25)$ | $1.142 \pm 0.026^{\dagger}$ | $13.78 \pm 0.33^{\dagger}$ | $-2.801 \pm 0.053$ | $-2.79 \pm 0.05$ |
| $\mathcal{L}_{\beta-\text{NLL}}(\beta=0.5)$ | $1.141 \pm 0.046^{\dagger}$ | $\mathbf{13.99 \pm 0.40}$ | $-2.805 \pm 0.052$ | $-2.78 \pm 0.02$ |
| $\mathcal{L}_{\beta-\text{NLL}}(\beta=0.75)$ | $1.137 \pm 0.041^{\dagger}$ | $13.63 \pm 0.62^{\dagger}$ | $-2.806 \pm 0.053$ | $-2.79 \pm 0.02$ |
| $\mathcal{L}_{\beta-\text{NLL}}(\beta=1.0)$ | $1.126 \pm 0.041$ | $13.59 \pm 0.30$ | $-2.810 \pm 0.051$ | $-2.80 \pm 0.03$ |
| $\mathcal{L}_{\text{MM}}$ | $0.999 \pm 0.063$ | $12.73 \pm 0.64$ | $-2.918 \pm 0.092$ | $-2.98 \pm 0.06$ |
| $\mathcal{L}_{\text{MSE}}$ | — | — | — | — |
| Student-t | $\mathbf{1.155 \pm 0.037}$ | $12.47 \pm 0.48$ | $\mathbf{-2.738 \pm 0.026}$ | $\mathbf{-2.56 \pm 0.02}$ |
| xVAMP | $1.147 \pm 0.037^{\dagger}$ | $12.44 \pm 0.60$ | $-2.788 \pm 0.032$ | $-2.74 \pm 0.03$ |
| xVAMP* | $1.147 \pm 0.036^{\dagger}$ | $12.80 \pm 0.55$ | $-2.785 \pm 0.036$ | $-2.71 \pm 0.02$ |
| VBEM | $1.019 \pm 0.054$ | $8.05 \pm 0.13$ | $-2.819 \pm 0.018$ | $-2.85 \pm 0.01$ |
| VBEM* | $1.138 \pm 0.036^{\dagger}$ | $13.10 \pm 0.47$ | $-2.783 \pm 0.031$ | $-2.73 \pm 0.01$ |

| | superconductivity (21263, 81, 1) | wine-red (1599, 11, 1) | wine-white (4898, 11, 1) | yacht (308, 6, 1) |
|---|---|---|---|---|
| $\mathcal{L}_{\beta-\text{NLL}}(\beta=0)$ | $-3.60 \pm 0.23$ | $-1.03 \pm 0.24^{\dagger}$ | $-1.059 \pm 0.074^{\dagger}$ | $-2.86 \pm 5.18$ |
| $\mathcal{L}_{\beta-\text{NLL}}(\beta=0.25)$ | $-3.56 \pm 0.14$ | $-0.98 \pm 0.12^{\dagger}$ | $-1.041 \pm 0.064^{\dagger}$ | $-1.97 \pm 1.14$ |
| $\mathcal{L}_{\beta-\text{NLL}}(\beta=0.5)$ | $-3.60 \pm 0.10$ | $-0.99 \pm 0.16^{\dagger}$ | $-1.036 \pm 0.065^{\dagger}$ | $-2.47 \pm 1.68$ |
| $\mathcal{L}_{\beta-\text{NLL}}(\beta=0.75)$ | $-3.72 \pm 0.11$ | $-1.02 \pm 0.22^{\dagger}$ | $-1.039 \pm 0.060^{\dagger}$ | $-1.87 \pm 0.55$ |
| $\mathcal{L}_{\beta-\text{NLL}}(\beta=1.0)$ | $-3.83 \pm 0.09$ | $-0.97 \pm 0.10^{\dagger}$ | $-1.040 \pm 0.067^{\dagger}$ | $-2.27 \pm 1.07$ |
| $\mathcal{L}_{\text{MM}}$ | $-4.45 \pm 0.39$ | $-1.22 \pm 0.43$ | $-1.135 \pm 0.093$ | $-11.24 \pm 31.03$ |
| $\mathcal{L}_{\text{MSE}}$ | — | — | — | — |
| Student-t | $\mathbf{-3.38 \pm 0.04}$ | $-0.94 \pm 0.10^{\dagger}$ | $-1.034 \pm 0.062^{\dagger}$ | $-1.23 \pm 0.55^{\dagger}$ |
| xVAMP | $-3.40 \pm 0.03$ | $-0.95 \pm 0.06^{\dagger}$ | $-1.038 \pm 0.050^{\dagger}$ | $-0.99 \pm 0.33^{\dagger}$ |
| xVAMP* | $-3.40 \pm 0.04^{\dagger}$ | $\mathbf{-0.94 \pm 0.06}$ | $-1.029 \pm 0.048^{\dagger}$ | $-1.04 \pm 0.47^{\dagger}$ |
| VBEM | $-3.63 \pm 0.09$ | $-0.95 \pm 0.07^{\dagger}$ | $\mathbf{-1.028 \pm 0.048}$ | $-2.65 \pm 0.10$ |
| VBEM* | $-3.40 \pm 0.04^{\dagger}$ | $-0.94 \pm 0.07^{\dagger}$ | $-1.031 \pm 0.057^{\dagger}$ | $\mathbf{-0.98 \pm 0.24}$ |

Table S2: Results for UCI Regression Datasets. RMSE (lower is better) and standard deviation, together with dataset size, input and output dimensions. Best mean value in bold. Results that are not statistically distinguishable from the best result are marked with †.

| | carbon (10721, 5, 3) | concrete (1030, 8, 1) | energy (768, 8, 2) | housing (506, 13, 1) |
|---|---|---|---|---|
| $\mathcal{L}_{\beta-\mathrm{NLL}}(\beta=0)$ | $0.0068 \pm 0.0029^\dagger$ | $6.08 \pm 0.65$ | $2.25 \pm 0.34$ | $3.56 \pm 1.07^\dagger$ |
| $\mathcal{L}_{\beta-\mathrm{NLL}}(\beta=0.25)$ | $0.0069 \pm 0.0028^\dagger$ | $5.79 \pm 0.74$ | $1.81 \pm 0.30$ | $3.48 \pm 1.15^\dagger$ |
| $\mathcal{L}_{\beta-\mathrm{NLL}}(\beta=0.5)$ | $0.0068 \pm 0.0029^\dagger$ | $5.61 \pm 0.65$ | $1.12 \pm 0.25$ | $3.42 \pm 1.04^\dagger$ |
| $\mathcal{L}_{\beta-\mathrm{NLL}}(\beta=0.75)$ | $0.0069 \pm 0.0028^\dagger$ | $5.67 \pm 0.73$ | $1.31 \pm 0.45$ | $3.43 \pm 1.07^\dagger$ |
| $\mathcal{L}_{\beta-\mathrm{NLL}}(\beta=1.0)$ | $0.0073 \pm 0.0026^\dagger$ | $5.55 \pm 0.77^\dagger$ | $1.54 \pm 0.54$ | $3.50 \pm 0.95^\dagger$ |
| $\mathcal{L}_{\mathrm{MM}}$ | $0.0097 \pm 0.0034$ | $6.28 \pm 0.82$ | $2.19 \pm 0.28$ | $4.02 \pm 1.18$ |
| $\mathcal{L}_{\mathrm{MSE}}$ | $0.0068 \pm 0.0028^\dagger$ | $\mathbf{4.96 \pm 0.64}$ | $\mathbf{0.92 \pm 0.11}$ | $3.24 \pm 1.08^\dagger$ |
| Student-t | $0.0067 \pm 0.0029^\dagger$ | $5.82 \pm 0.59$ | $2.26 \pm 0.34$ | $3.48 \pm 1.17^\dagger$ |
| xVAMP | $0.0067 \pm 0.0029^\dagger$ | $5.44 \pm 0.64^\dagger$ | $1.87 \pm 0.32$ | $3.23 \pm 1.00^\dagger$ |
| xVAMP* | $0.0067 \pm 0.0029^\dagger$ | $5.35 \pm 0.73^\dagger$ | $2.00 \pm 0.26$ | $3.38 \pm 1.15^\dagger$ |
| VBEM | $0.0074 \pm 0.0026^\dagger$ | $5.21 \pm 0.58^\dagger$ | $1.29 \pm 0.33$ | $3.32 \pm 1.06^\dagger$ |
| VBEM* | $\mathbf{0.0067 \pm 0.0029}$ | $5.17 \pm 0.59^\dagger$ | $1.08 \pm 0.17$ | $\mathbf{3.19 \pm 1.02}$ |

| | kin8m (8192, 8, 1) | naval (11934, 16, 2) | power (9568, 4, 1) | protein (45730, 9, 1) |
|---|---|---|---|---|
| $\mathcal{L}_{\beta-\mathrm{NLL}}(\beta=0)$ | $0.087 \pm 0.004$ | $0.0021 \pm 0.0006$ | $4.06 \pm 0.18^\dagger$ | $4.49 \pm 0.11$ |
| $\mathcal{L}_{\beta-\mathrm{NLL}}(\beta=0.25)$ | $0.083 \pm 0.003$ | $0.0012 \pm 0.0004$ | $4.04 \pm 0.18^\dagger$ | $4.35 \pm 0.05^\dagger$ |
| $\mathcal{L}_{\beta-\mathrm{NLL}}(\beta=0.5)$ | $0.082 \pm 0.003^\dagger$ | $0.0006 \pm 0.0002$ | $4.04 \pm 0.17^\dagger$ | $4.31 \pm 0.02^\dagger$ |
| $\mathcal{L}_{\beta-\mathrm{NLL}}(\beta=0.75)$ | $0.081 \pm 0.004^\dagger$ | $0.0004 \pm 0.0001^\dagger$ | $4.04 \pm 0.15^\dagger$ | $4.28 \pm 0.02^\dagger$ |
| $\mathcal{L}_{\beta-\mathrm{NLL}}(\beta=1.0)$ | $0.081 \pm 0.003^\dagger$ | $\mathbf{0.0004 \pm 0.0000}$ | $4.06 \pm 0.18^\dagger$ | $4.31 \pm 0.05^\dagger$ |
| $\mathcal{L}_{\mathrm{MM}}$ | $0.082 \pm 0.003^\dagger$ | $0.0005 \pm 0.0001$ | $4.07 \pm 0.16^\dagger$ | $4.32 \pm 0.07^\dagger$ |
| $\mathcal{L}_{\mathrm{MSE}}$ | $\mathbf{0.081 \pm 0.003}$ | $0.0004 \pm 0.0001^\dagger$ | $\mathbf{4.01 \pm 0.19}$ | $\mathbf{4.28 \pm 0.07}$ |
| Student-t | $0.085 \pm 0.005$ | $0.0026 \pm 0.0009$ | $4.02 \pm 0.16^\dagger$ | $4.76 \pm 0.24$ |
| xVAMP | $0.081 \pm 0.003^\dagger$ | $0.0023 \pm 0.0004$ | $4.03 \pm 0.17^\dagger$ | $4.38 \pm 0.05^\dagger$ |
| xVAMP* | $0.082 \pm 0.003^\dagger$ | $0.0020 \pm 0.0006$ | $4.03 \pm 0.18^\dagger$ | $4.31 \pm 0.02^\dagger$ |
| VBEM | $0.082 \pm 0.003^\dagger$ | $0.0009 \pm 0.0004$ | $4.09 \pm 0.15^\dagger$ | $4.31 \pm 0.01^\dagger$ |
| VBEM* | $0.082 \pm 0.004^\dagger$ | $0.0015 \pm 0.0005$ | $4.02 \pm 0.18^\dagger$ | $4.35 \pm 0.09^\dagger$ |

| | superconductivity (21263, 81, 1) | wine-red (1599, 11, 1) | wine-white (4898, 11, 1) | yacht (308, 6, 1) |
|---|---|---|---|---|
| $\mathcal{L}_{\beta-\mathrm{NLL}}(\beta=0)$ | $13.87 \pm 0.50$ | $0.636 \pm 0.038^\dagger$ | $0.691 \pm 0.032^\dagger$ | $1.22 \pm 0.47$ |
| $\mathcal{L}_{\beta-\mathrm{NLL}}(\beta=0.25)$ | $13.50 \pm 0.49$ | $0.638 \pm 0.036^\dagger$ | $0.687 \pm 0.039^\dagger$ | $1.73 \pm 1.00$ |
| $\mathcal{L}_{\beta-\mathrm{NLL}}(\beta=0.5)$ | $13.02 \pm 0.47$ | $0.635 \pm 0.037^\dagger$ | $0.685 \pm 0.035^\dagger$ | $2.35 \pm 1.44$ |
| $\mathcal{L}_{\beta-\mathrm{NLL}}(\beta=0.75)$ | $13.20 \pm 0.46$ | $0.638 \pm 0.035^\dagger$ | $0.689 \pm 0.034^\dagger$ | $1.97 \pm 1.03$ |
| $\mathcal{L}_{\beta-\mathrm{NLL}}(\beta=1.0)$ | $13.42 \pm 0.41$ | $0.639 \pm 0.035^\dagger$ | $0.684 \pm 0.031^\dagger$ | $2.08 \pm 1.13$ |
| $\mathcal{L}_{\mathrm{MM}}$ | $13.68 \pm 0.79$ | $0.652 \pm 0.044^\dagger$ | $0.692 \pm 0.032^\dagger$ | $3.02 \pm 1.38$ |
| $\mathcal{L}_{\mathrm{MSE}}$ | $\mathbf{12.48 \pm 0.40}$ | $0.633 \pm 0.036^\dagger$ | $\mathbf{0.684 \pm 0.038}$ | $0.78 \pm 0.25^\dagger$ |
| Student-t | $13.52 \pm 0.60$ | $0.636 \pm 0.038^\dagger$ | $0.688 \pm 0.036^\dagger$ | $1.34 \pm 0.63$ |
| xVAMP | $13.33 \pm 0.52$ | $0.635 \pm 0.035^\dagger$ | $0.691 \pm 0.032^\dagger$ | $0.99 \pm 0.43$ |
| xVAMP* | $13.42 \pm 0.59$ | $0.633 \pm 0.035^\dagger$ | $0.685 \pm 0.032^\dagger$ | $1.13 \pm 0.66$ |
| VBEM | $12.72 \pm 0.57^\dagger$ | $0.639 \pm 0.041^\dagger$ | $0.685 \pm 0.035^\dagger$ | $1.66 \pm 0.84$ |
| VBEM* | $13.15 \pm 0.43$ | $\mathbf{0.633 \pm 0.040}$ | $0.686 \pm 0.036^\dagger$ | $\mathbf{0.65 \pm 0.20}$ |

shows qualitative examples. Our $\beta-$NLL loss with $\beta > 0$ allows to learn good reconstructions and meaningful uncertainties. Moreover, images produced by sampling latents from the prior are semantically meaningful, indicating that adding our loss function does not break the disentangling properties of VAEs. Compare that to Student-t, xVAMP(*), and VBEM(*), which do not produce similarly clear images when sampling from the prior.

Table S3: Results for generative modeling with variational autoencoders on MNIST and Fashion-MNIST. We report RMSE and posterior predictive log-likelihood (LL) with standard deviation over 5 independent trials.

| | MNIST | | Fashion-MNIST | |
|---|---|---|---|---|
| Loss | RMSE ↓ | LL ↑ | RMSE ↓ | LL ↑ |
| $\mathcal{L}_{\mathrm{NLL}}\,(\sigma^2 = 1)$ | $0.153 \pm 0.002$ | $-730 \pm 0$ | $0.143 \pm 0.002$ | $-729 \pm 0$ |
| $\mathcal{L}_{\beta-\mathrm{NLL}}(\beta = 0)$ | $0.237 \pm 0.002$ | $2116 \pm 55$ | $0.170 \pm 0.001$ | $1940 \pm 104$ |
| $\mathcal{L}_{\beta-\mathrm{NLL}}(\beta = 0.25)$ | $0.181 \pm 0.004$ | $2511 \pm 88$ | $0.140 \pm 0.002$ | $2010 \pm 39$ |
| $\mathcal{L}_{\beta-\mathrm{NLL}}(\beta = 0.5)$ | $0.151 \pm 0.003$ | $2220 \pm 25$ | $0.125 \pm 0.003$ | $1639 \pm 52$ |
| $\mathcal{L}_{\beta-\mathrm{NLL}}(\beta = 0.75)$ | $0.142 \pm 0.001$ | $1954 \pm 50$ | $0.131 \pm 0.001$ | $1331 \pm 32$ |
| $\mathcal{L}_{\beta-\mathrm{NLL}}(\beta = 1.0)$ | $0.152 \pm 0.001$ | $1706 \pm 30$ | $0.138 \pm 0.002$ | $1142 \pm 26$ |
| $\mathcal{L}_{\mathrm{MM}}$ | $0.260 \pm 0.000$ | $385 \pm 11$ | $0.295 \pm 0.000$ | $-151 \pm 4$ |
| Student-t | $0.273 \pm 0.002$ | $4291 \pm 103$ | $0.182 \pm 0.002$ | $2857 \pm 9$ |
| xVAMP | $0.225 \pm 0.002$ | $2989 \pm 268$ | $0.161 \pm 0.001$ | $2158 \pm 100$ |
| xVAMP* | $0.225 \pm 0.001$ | $3062 \pm 215$ | $0.160 \pm 0.002$ | $2150 \pm 131$ |
| VBEM | $0.114 \pm 0.001$ | $719 \pm 9$ | $0.108 \pm 0.000$ | $660 \pm 2$ |
| VBEM* | $0.176 \pm 0.008$ | $3213 \pm 238$ | $0.150 \pm 0.003$ | $2244 \pm 78$ |

## B.6 DEPTH REGRESSION

We evaluate $\beta-$NLL on the task of depth regression on the NYUv2 dataset (Silberman et al., 2012). For this purpose, we use a state-of-the-art method for depth regression, AdaBins (Bhat et al., 2021), and train it with different loss functions. Note that we remove the Mini-ViT Transformer module from the model, thus our results are not directly comparable with those reported by Bhat et al. (2021).

Table S4 presents quantitative results. $\beta-$NLL with $\beta > 0$ achieves better RMSE than the NLL loss. $\beta-$NLL with $\beta = 0.5$ again provides a good trade-off, achieving similar RMSE as $\mathcal{L}_{\mathrm{MSE}}$ and performing better on some of the other metrics. Figure S6 shows qualitative examples. Compared to $\mathcal{L}_{\mathrm{NLL}}$, the depth maps predicted by $\beta-$NLL with $\beta > 0$ are noticeably sharper.

Table S4: Results for depth regression on the NYUv2 dataset (Silberman et al., 2012). We adapt a state-of-the-art network for depth regression from AdaBins (Bhat et al., 2021) and train it with different loss functions. In contrast to AdaBins, our network does not include the Mini-ViT Transformer module and thus our results are not directly comparable with those originally reported. For reference, we also report numbers from other recent literature on this task. Notably, the Gaussian NLL in all variants clearly outperforms the Scale Invariant (SI) loss, despite the latter being a loss function specifically designed for the task of depth regression. We refer the reader to Bhat et al. (2021) for a description of the metrics.

| Method | $\delta_1$ ↑ | $\delta_2$ ↑ | $\delta_3$ ↑ | REL ↓ | RMSE ↓ | $\log_{10}$ ↓ | LL ↑ |
|---|---|---|---|---|---|---|---|
| $\mathcal{L}_{\beta-\mathrm{NLL}}(\beta = 0)$ | 0.8855 | 0.9796 | 0.9959 | 0.1094 | 0.3854 | 0.0462 | -4.52 |
| $\mathcal{L}_{\beta-\mathrm{NLL}}(\beta = 0.25)$ | 0.8887 | 0.9812 | 0.9956 | 0.1081 | 0.3818 | 0.0458 | -8.15 |
| $\mathcal{L}_{\beta-\mathrm{NLL}}(\beta = 0.5)$ | 0.8885 | 0.9813 | 0.9956 | 0.1093 | 0.3789 | 0.0458 | -7.50 |
| $\mathcal{L}_{\beta-\mathrm{NLL}}(\beta = 0.75)$ | 0.8902 | 0.9804 | 0.9952 | 0.1095 | 0.3800 | 0.0462 | -7.35 |
| $\mathcal{L}_{\beta-\mathrm{NLL}}(\beta = 1.0)$ | 0.8872 | 0.9813 | 0.9958 | 0.1088 | 0.3845 | 0.0467 | -5.10 |
| $\mathcal{L}_{\mathrm{MSE}}$ | 0.8890 | 0.9806 | 0.9960 | 0.1086 | 0.3776 | 0.0461 | — |
| $\mathcal{L}_1$ | 0.8877 | 0.9798 | 0.9955 | 0.1073 | 0.3850 | 0.0459 | — |
| SI Loss (Bhat et al., 2021) | 0.881 | 0.980 | 0.996 | 0.111 | 0.419 | — | — |
| AdaBins (Bhat et al., 2021) | 0.903 | 0.984 | 0.997 | 0.103 | 0.364 | 0.044 | — |
| BTS (Lee et al., 2019) | 0.885 | 0.978 | 0.994 | 0.110 | 0.392 | 0.047 | — |
| DAV (Chen et al., 2019) | 0.882 | 0.980 | 0.996 | 0.108 | 0.412 | — | — |

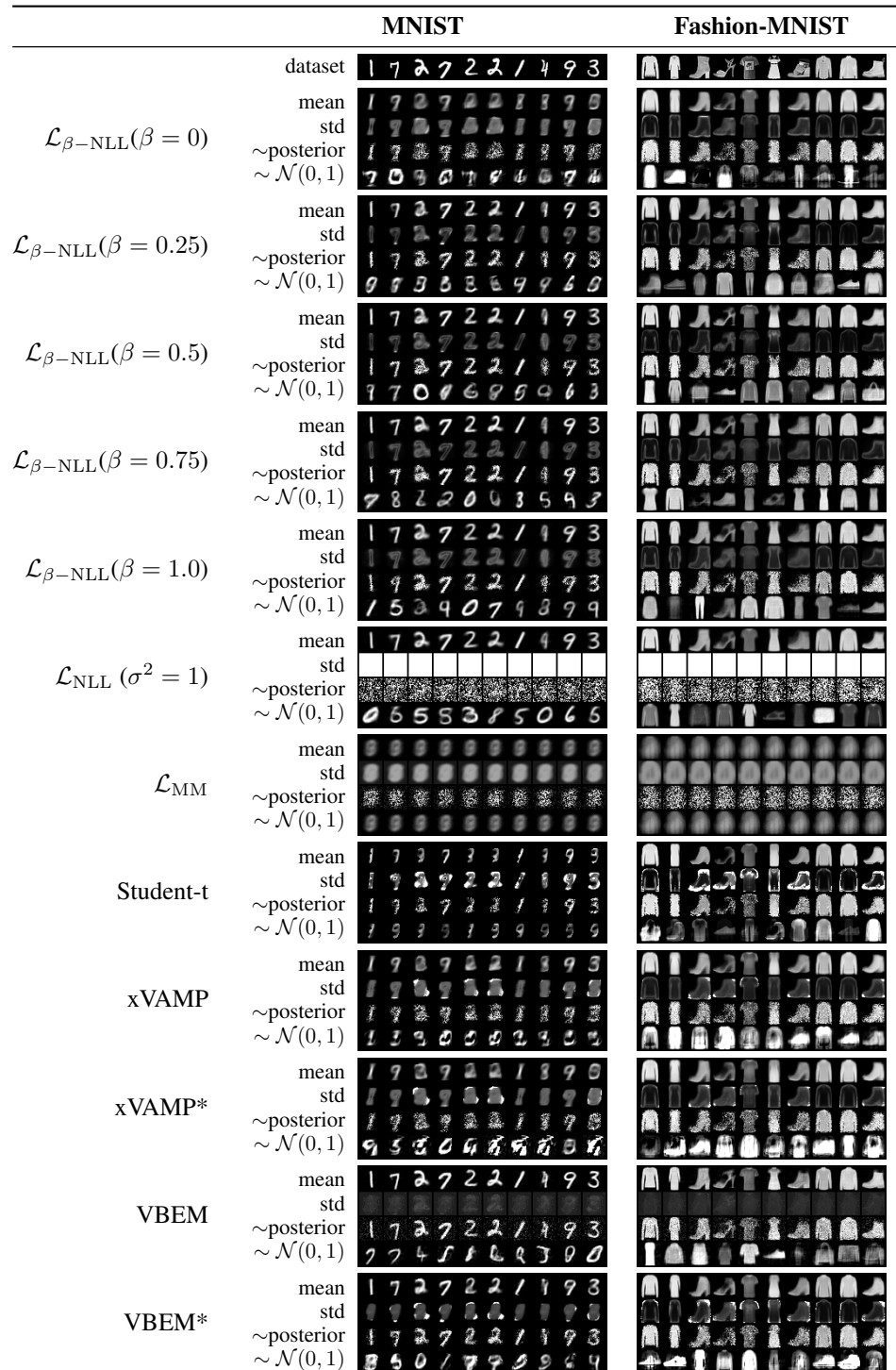

Figure S5: Generative modeling with variational autoencoders on MNIST and Fashion-MNIST. The overall first row shows inputs from the test set. For each method, we present posterior predictive means (i.e. reconstructions), the posterior predictive standard deviations, samples from the posterior predictive distribution, and finally samples using the prior $\mathcal{N}(0, 1)$. $\mathcal{L}_{\mathrm{NLL}}(\sigma^2 = 1)$ refers to Gaussian log-likelihood with a fixed variance of 1. Values are clipped to the interval $[0, 1]$. Examples are not cherry-picked.

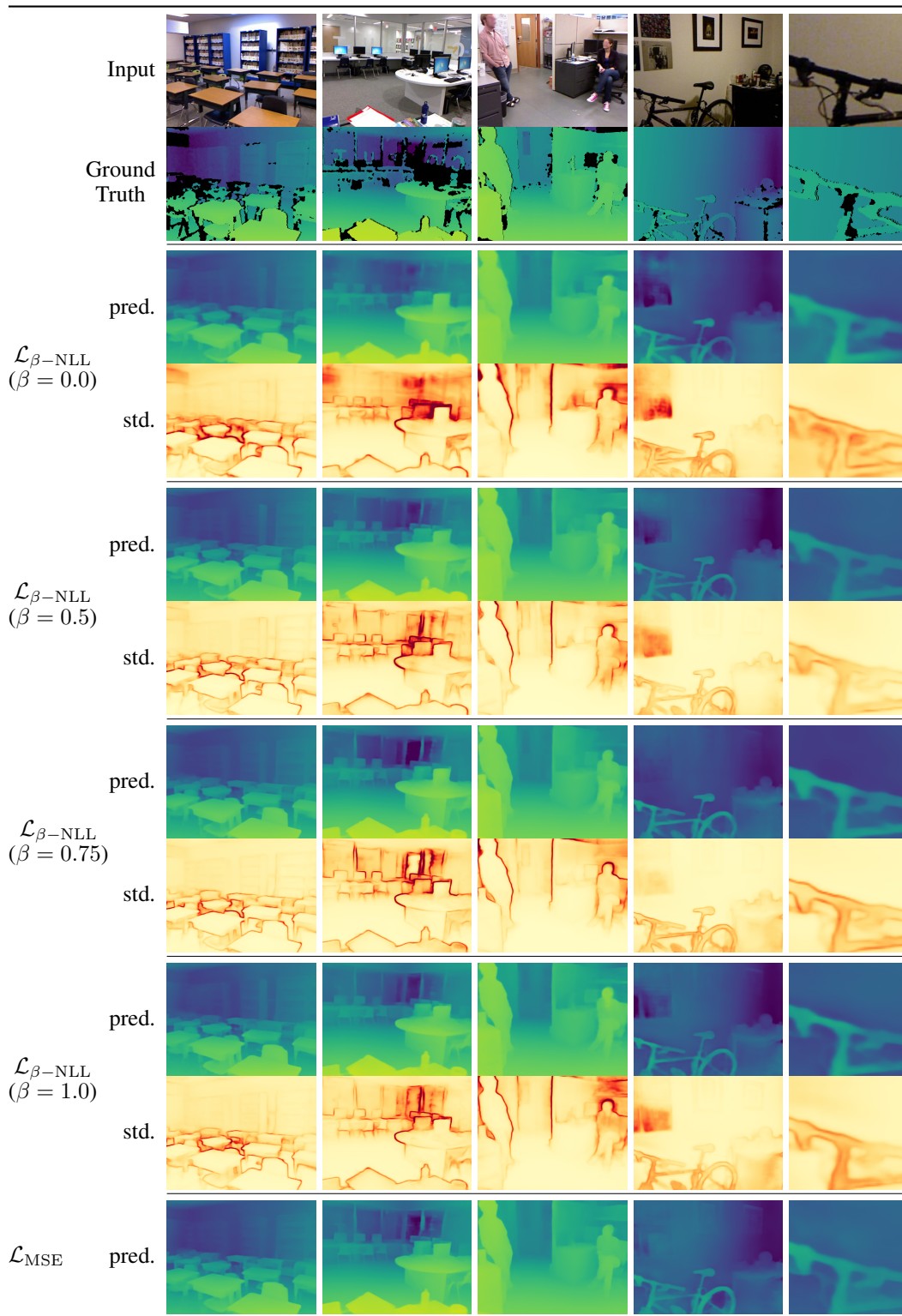

Figure S6: Example results for depth regression on the NYUv2 dataset (Silberman et al., 2012). First two rows show input image and ground truth depth map, where black values in the depth map represent missing values. For each method, we present the predicted depth map (pred.), and the aleatoric uncertainty in form of the predicted standard deviation (std.). Results for $\mathcal{L}_{\beta-\text{NLL}}$ with $\beta > 0$ are noticeably sharper. The last column shows a magnified view on the previous picture.

## C   DATASETS AND TRAINING SETTINGS

**Sinusoidal without heteroscedastic noise**   This dataset is created by taking $1\,000$ uniformly spaced points on the interval $[0, 12]$ as inputs $x$ and applying the function $y(x) = 0.4 \sin(2\pi x) + \xi$ to them to create the targets $y$, where $\xi$ is Gaussian noise with a standard deviation of $0.01$.

**Sinusoidal with heteroscedastic noise**   We use the synthetic data as introduced in Detlefsen et al. (2019). From the functional form $y = x \sin(x) + x\xi_1 + \xi_2$, with Gaussian noise with standard deviation $\sigma = 0.3$ for $\xi_1$ and $\xi_2$, we sample $500$ points uniformly spaced in the interval $[0, 10]$. The model is an MLP with one hidden layer of $50$ units and $\texttt{tanh}$ activations (as used in Detlefsen et al. (2019) and Stirn & Knowles (2020)).

**UCI Datasets**   We use the UCI datasets suite commonly used to benchmark uncertainty estimation, stemming from the UCI Machine Learning Repository.[3] In particular, we use the training-test protocol from (Hernández-Lobato & Adams, 2015; Gal & Ghahramani, 2016), and their data splits[4], except for "carbon", "energy", "naval", "superconductivity", and "wine-white", where we generate our own random splits.

Inputs and targets are whitened on the training set. Metrics are reported in the original scale of the data. Each dataset is divided into 20 randomly sampled train-test splits (80%-20%). For each split, we further randomly divide the training set into 80% training data and 20% validation data and search for an optimal learning rate from the set $\{10^{-4}, 3 \cdot 10^{-4}, 7 \cdot 10^{-4}, 10^{-3}, 3 \cdot 10^{-3}, 7 \cdot 10^{-3}\}$ by monitoring log-likelihood on the validation set. We train for a maximum of $20\,000$ updates, except for the larger "kin8m", "power plant", "protein", and "naval" datasets where we train for a maximum of $100\,000$ updates. We perform early-stopping with a patience of 50 epochs, retrain the model with the best found learning rate on the full training set, and then evaluate on the test split. The reported performance and standard deviations are taken as averages over all test splits. Note that the performance we report is not comparable with other publications, as performance is known to differ strongly over different data splits. Some other works also perform early-stopping on the test set, which distorts the results.

We use a single-layer $\texttt{relu}$ hidden network with 50 neurons, except for "protein" where we use 100 neurons. The batch size is 256.

Following Stirn & Knowles (2020), for each method, we report the number of datasets for which the method is statistically indistinguishable from the respective best method in Table 1 (*Ties*). For this purpose, we performed a two-sided Kolmogorov-Smirnov test with a $p \leq 0.05$ significance level.

**ObjectSlide**   This environment consists of an agent whose task is to slide an object to a target location (Seitzer et al., 2021). The continuous state space consists of 4 dimensions: agent and object positions and velocities, and the continuous action space is a one-dimensional movement command. The forward prediction task consists of predicting the change in object position in the next state from the current state and action. The dataset we use consists of $180\,000$ transitions collected using a random policy, which we split into training, validation, and testing sets with $60\,000$ transitions each. Inputs and targets are whitened on the training set. Metrics are reported in the original scale of the data. We train for a maximum of $5\,000$ epochs with a batch size of 256, and evaluate the model with the best validation log-likelihood on the test set afterwards.

**Fetch-PickAndPlace**   We use the Fetch-PickAndPlace environment (Plappert et al., 2018) from OpenAI Gym (Brockman et al., 2016) as a challenging real-world scenario. The task of the agent is to use a position-controlled 7 DoF robotic arm to lift an object to a target location in space. The state space is 25-dimensional and the action space is 4-dimensional. As in ObjectSlide, the prediction task is to predict the 3-dimensional change in object position from the current state and action. We use $840\,000$ transitions collected using the APEX method (Pinneri et al., 2021) as our dataset, which we split into 70% training, 15% validation, and 15% testing data. Inputs and targets are whitened on the training set. Metrics are reported in the original scale of the data. We train for a maximum of 500

---

[3]https://archive.ics.uci.edu
[4]available under https://github.com/yaringal/DropoutUncertaintyExps

epochs with a batch size of 256, and evaluate the model with the best validation log-likelihood on the test set afterwards.

**NYUv2 Depth Regression**  We use the dataset in the variant provided by Lee et al. (2019).[5] Training settings and evaluation protocol were taken from Bhat et al. (2021). We train for 25 epochs using a batch size of 16 and validate the model every 100 updates. We use the model with the best "REL" metric on the validation set for testing.

## D  HYPERPARAMETER SETTINGS AND IMPLEMENTATION DETAILS

For all experiments, we used the Adam optimizer (Kingma & Ba, 2015) with standard settings $\beta_1 = 0.9, \beta_2 = 0.999$. We parameterize the Gaussian distribution using two linear layers on top of shared features produced by an MLP. The variance $\hat{\sigma}^2(x)$ is constrained to the positive region using the $\mathrm{softplus}(x) = \log(1 + \exp(x))$ activation function. We additionally add a small constant of $10^{-8}$ to prevent the variance from collapsing to zero and clamp the maximum variance to $1\,000$.

Some baselines use a Student's t-distribution (Student's t, xVAMP(*), VBEM(*)) as their predictive distribution. This distribution results from integrating out the unknown variance of a Gaussian with a learned Gamma prior on the inverse variance (Detlefsen et al., 2019; Stirn & Knowles, 2020). We parametrize the Gamma distribution in terms of data-dependent alpha and beta parameters, i.e. $\alpha(x)$ and $\beta(x)$, which are computed using linear layers on top of the shared features. In this case, the MLP has three outputs: mean, alpha, and beta. Both alpha and beta are constrained to the positive region using the $\mathrm{softplus}$ activation. We add a positive constant of $1.001$ for alpha and $10^{-8} \cdot 0.001$ for beta. Alpha is clamped to a maximum value of $1\,000$ and beta to $10^{-8} \cdot 999$. These values were chosen such that the resulting variance matches the range $(10^{-8}, 1000]$ while ensuring that the degrees-of-freedom parameter $\nu$ of the Student's t is always greater than 2.

For xVAMP and VBEM, the MLP outputs a fourth term, $\pi(x)$, representing the logits of a categorical distribution that specifies the mixture weights of the prior. We initialize the prior parameters exactly the same as Stirn & Knowles (2020). For these methods, the objective function additionally contains a KL divergence between a Gamma distribution and a mixture-of-Gamma distributions. Following Stirn & Knowles (2020), we approximate this KL divergence using 20 Monte-Carlo samples.

### D.1  SINUSOIDAL REGRESSION PROBLEM

The sinusoidal fit in Fig. 1 results from a network of two hidden layers with 128 neurons per layer and $\mathrm{tanh}$ activations, optimized with a learning rate of $5 \cdot 10^{-4}$ and a batch size of 100. For the experiment in Sec. 5.1, we scan over learning rates and architectures with different hidden layers and units per layer, as detailed in Table S5.

Table S5: Architectures used for the sinusoidal regression task. Fully-connected feed-forward neural networks with $\mathrm{tanh}$ activations.

| Architecture # | 0 | 1 | 2 | 3 | 4 |
|---|---|---|---|---|---|
| # Hidden Layers | 2 | 2 | 2 | 3 | 3 |
| # Units per Layer | 32 | 64 | 128 | 128 | 256 |

### D.2  OBJECTSLIDE AND FETCH-PICKANDPLACE

For each tested loss function, we performed a grid search on the ObjectSlide and Fetch-PickAndPlace datasets. We report the parameters we scanned over in Table S6. Table S7 reports the model configurations with the best validation log-likelihood on the grid search. For the results in Table 2, we retrained the best model configuration with five different random seeds and evaluated them on the hold-out test set.

---

[5]available under `https://github.com/cogaplex-bts/bts`

Table S6: Hyperparameter settings for our grid search on the ObjectSlide and Fetch-PickAndPlace datasets. We run 96 configurations per loss function.

| Hyperparameter | Set of Values |
|---|---|
| Learning Rate | $\{3 \cdot 10^{-5}, 10^{-4}, 3 \cdot 10^{-4}, \cdot 10^{-3}\}$ |
| # Hidden Layers | $\{2, 3, 4\}$ |
| # Units per Layer | $\{128, 256, 386, 512\}$ |
| Activation | $\{\mathrm{tanh}, \mathrm{relu}\}$ |

Table S7: Best hyperparameters found by grid search on ObjectSlide and Fetch-PickAndPlace datasets, measured by best log-likelihood on the validation set.

(a) ObjectSlide

| Method | $\beta$ | LR | Layers | Act. |
|---|---|---|---|---|
| $\mathcal{L}_{\mathrm{MSE}}$ | | $10^{-3}$ | $3 \times 128$ | relu |
| $\mathcal{L}_{\mathrm{NLL}}$ | | $10^{-3}$ | $3 \times 128$ | relu |
| $\mathcal{L}_{\beta-\mathrm{NLL}}$ | 0.25 | $10^{-3}$ | $3 \times 128$ | relu |
| $\mathcal{L}_{\beta-\mathrm{NLL}}$ | 0.5 | $10^{-3}$ | $3 \times 128$ | relu |
| $\mathcal{L}_{\beta-\mathrm{NLL}}$ | 0.75 | $10^{-3}$ | $3 \times 128$ | relu |
| $\mathcal{L}_{\beta-\mathrm{NLL}}$ | 1.0 | $10^{-3}$ | $3 \times 128$ | relu |
| $\mathcal{L}_{\mathrm{MM}}$ | | $10^{-3}$ | $3 \times 128$ | relu |
| Student-t | | $10^{-3}$ | $2 \times 386$ | relu |
| xVAMP | | $10^{-4}$ | $4 \times 128$ | relu |
| xVAMP* | | $10^{-4}$ | $3 \times 256$ | relu |
| VBEM | | $3 \cdot 10^{-4}$ | $2 \times 256$ | tanh |
| VBEM* | | $10^{-3}$ | $2 \times 386$ | relu |

(b) Fetch-PickAndPlace

| Method | $\beta$ | LR | Layers | Act. |
|---|---|---|---|---|
| $\mathcal{L}_{\mathrm{MSE}}$ | | $10^{-3}$ | $4 \times 128$ | relu |
| $\mathcal{L}_{\mathrm{NLL}}$ | | $3 \cdot 10^{-4}$ | $4 \times 128$ | relu |
| $\mathcal{L}_{\beta-\mathrm{NLL}}$ | 0.25 | $3 \cdot 10^{-4}$ | $4 \times 128$ | relu |
| $\mathcal{L}_{\beta-\mathrm{NLL}}$ | 0.5 | $3 \cdot 10^{-4}$ | $4 \times 128$ | relu |
| $\mathcal{L}_{\beta-\mathrm{NLL}}$ | 0.75 | $10^{-3}$ | $4 \times 128$ | relu |
| $\mathcal{L}_{\beta-\mathrm{NLL}}$ | 1.0 | $10^{-3}$ | $4 \times 128$ | relu |
| $\mathcal{L}_{\mathrm{MM}}$ | | $10^{-3}$ | $4 \times 128$ | relu |
| Student-t | | $3 \cdot 10^{-4}$ | $3 \times 256$ | relu |
| xVAMP | | $10^{-4}$ | $3 \times 386$ | relu |
| xVAMP* | | $10^{-4}$ | $3 \times 386$ | relu |
| VBEM | | $10^{-3}$ | $3 \times 386$ | relu |
| VBEM* | | $10^{-4}$ | $3 \times 386$ | relu |

### D.3 VARIATIONAL AUTOENCODERS

We largely follow Stirn & Knowles (2020) for their training settings for the VAE experiment. In particular, we use an encoder with three layers of $512, 256, 128$ neurons and a decoder with three layers of $128, 256, 512$ neurons, all with relu activations. The latent space is 10-dimensional for MNIST and 25-dimensional for FashionMNIST. We train the VAEs for a maximum of $1\,000$ epochs, using Adam with a learning rate of $0.0003$ and a batch size of $256$. Early-stopping with a patience of $50$ epochs is performed on the log-likelihood of the validation set. The validation set consists of 20% of the MNIST/FashionMNIST training set.

### D.4 DEPTH REGRESSION

We use the official implementation of AdaBins (Bhat et al., 2021),[6] thereby reproducing their exact training settings and evaluation protocol. We remove the AdaBins/mini-ViT Transformer from the model. Instead, the feature map output by the U-Net is reduced to two channels using a $1 \times 1$ convolution, where we use the first channel as the mean predictor and the second channel as the variance predictor. In this setting, both mean and variance are constrained to positive numbers by a softplus activation. On top of that, we add a positive offset to ensure a minimum output value of $10^{-3}$ for the mean (the minimum possible depth value) and $10^{-6}$ for the variance and clamp both mean and variance to a maximum value of $10$.

---

[6]https://github.com/shariqfarooq123/AdaBins

### D.5 Implementation of beta-NLL in Pytorch

```python
def beta_nll_loss(mean, variance, target, beta):
    """Compute beta-NLL loss

    :param mean: Predicted mean of shape B x D
    :param variance: Predicted variance of shape B x D
    :param target: Target of shape B x D
    :param beta: Parameter from range [0, 1] controlling relative
        weighting between data points, where '0' corresponds to
        high weight on low error points and '1' to an equal weighting.
    :returns: Loss per batch element of shape B
    """
    loss = 0.5 * ((target - mean) ** 2 / variance + variance.log())

    if beta > 0:
        loss = loss * variance.detach() ** beta

    return loss.sum(axis=-1)
```

