# OpenReview forum: "On the Pitfalls of Heteroscedastic Uncertainty Estimation with Probabilistic Neural Networks"
_ICLR.cc/2022/Conference — ICLR 2022 Poster_

### Official Review · Reviewer_gjks · 2021-11-01

**Correctness:** 4
**Technical Novelty And Significance:** 3
**Empirical Novelty And Significance:** 3
**Recommendation:** 6
**Confidence:** 4

**Main Review:**

The problem of optimizing Gaussian likelihoods is ubiquitous and, thus, understanding its pitfalls thoroughly is important. While the phenomenon of variance weighting is known, the authors provide a nice analysis with explanations, visualizations, and derivations of  high quality. The claims they make in Secs. 3 and 4 are all well supported and fleshed out nicely. The presentation is very clear and well-organized. I really enjoyed reading the paper and found it quite insightful. The authors provide a pragmatic and easy-to-implement method to alleviate the identified problems in a situation-specific manner (by adjusting $\beta$).

Unfortunately, I feel the experimental evaluation is lacking:
- The results in Tab. 1 and Tab. S1 seem to be not significant according to the reported standard errors? How many seeds were evaluated? Tab. S1 should be moved to the main part (assessing the MSE values is central to judging the effectiveness of the proposed method).
- Why is the RMSE of MM in Tab. 1 worse than that of $\beta$-NLL? It is optimizing directly for the mean and thus should yield better results, shouldn't it? Did you use two-headed networks? If so, this might explain the worse MSE of MM. But then, I'm still wondering why the MSE in the dynamics experiments (Tab. 2) is worse than $\beta$-NLL as here you train on the pure MSE loss (not MM). Why don't you evaluate pure MSE on the experiments in Tab. 1/S1?
- While I agree that the proposed method visually seems to yield better calibrated uncertainty estimates than MM in Fig. 8, I would also like to see a quantitative evaluation here.
- The tasks are low-dimensional (1D, 3D). I would be interested in results on more complex, higher dimensional problems.

Minor points not influencing my score:

- Shouldn't Eq. 6 contain a factor of $Z$?
- Sec. 5.1. misses a reference to Fig. 7.
- On p. 8, paragraph "Sinusoidal with heteroscedastic noise", $\xi$ should read $\xi_1$ and $\xi_2$.
- Fig. 9: $\beta$ is not rendered in the labels.
- Sec. 6: wrong full stop before "we reveal the underlying reasons".

**Summary Of The Paper:**

The authors analyze the known phenomenon that the optimization of the Gaussian likelihood of a probabilistic neural network with heteroscedastic output variance using gradient ascent can get stuck at solutions with a suboptimal mean fit, compensated by large output variance: the likelihood amplifies non-uniform initial distributions of feature granularity due to variance weighting, i.e., the gradient w.r.t. the mean are scaled by the inverse variance leading to well-fitted samples dominating the gradient. The authors argue that this effect can be undesirable because it can prevent spending model expressivity on hard-to-fit regions, while it can be desirable in other situations because it allows the network to ignore outliers arising due to noise. This is complementary to the MSE objective which is dominated by badly-fitted samples (and thus hard to fit regions) but sensitive to outliers, and which does not allow to fit the output variance. The authors propose a modified loss function, $\beta$-NLL, which interpolates between the likelihood and the MSE loss functions with the incentive to retain the "best of both worlds". They compare $\beta$-NLL with standard NLL, MSE, and a moment-matching version on several datasets.

**Summary Of The Review:**

The authors provide a very nice discussion and analysis of the known phenomenon of variance weighting arising in SGD on NLL loss of Gaussian likelihoods. While I judge this to be an interesting contribution on its own, I would like to see an improved experimental evaluation. Indeed, I am not yet fully convinced that the proposed $\beta$-NLL is effective. In its current state I rate the submission as borderline but I will raise my score if the authors improve upon the mentioned points during the rebuttal.

------------------------------

**Update after rebuttal:**
I thank the authors for their effort to improve the experimental evaluation. I raise my score to 6.

---

> ### Author Response · Authors · 2021-11-18
> **Response to Review**
>
> Thank you for your constructive and detailed review. We are glad that you enjoyed reading the paper and found it insightful. Below we address your questions / concerns.
>
> > The results in Tab. 1 and Tab. S1 seem to be not significant according to the reported standard errors. How many seeds were evaluated?
>
> The performance is evaluated using cross validation over 20 random different splits. Mean and standard deviation are taken over those splits. Additionally, we now report whether results of a method are statistically distinguishable from the best result on a dataset (see “Ties” in Tab. 1). We used a two-sided Kolmogorov-Smirnov test for this purpose.
>
> > Tab. S1 should be moved to the main part (assessing the MSE values is central to judging the effectiveness of the proposed method).
>
> Thank you for this suggestion. Due to space constraints, we have opted for a compromise. Table 1 now shows both log-likelihood and RMSE for 4 of the UCI datasets, but reports the *Ties* over all 12 datasets. The full results are included in tables S1 and S2 in the appendix.
>
> > Why is the RMSE of MM in Tab. 1 worse than that of $\beta$-NLL? It is optimizing directly for the mean and thus should yield better results, shouldn't it? Did you use two-headed networks? If so, this might explain the worse MSE of MM. But then, I'm still wondering why the MSE in the dynamics experiments (Tab. 2) is worse than $\beta$-NLL as here you train on the pure MSE loss (not MM). Why don't you evaluate pure MSE on the experiments in Tab. 1/S1?
>
> Yes, we use two-headed networks throughout the whole paper. We believe the reason MM is worse is that it sometimes exhibits instability, stemming from the loss on the variance. As for why the $\beta$-NLL loss on the dynamics experiments does perform (slightly) better than the MSE loss, we conjecture that this is due to a positive influence of the variance loss term on training the shared features, but we have not analysed this further.
>
> We did not evaluate pure MSE for the UCI datasets as it does not provide log-likelihood, and so did not match with the evaluation protocol for UCI, which uses early stopping on the log-likelihood. But we agree that it adds interesting information, and have also ran the MSE loss, using early stopping on MSE for the updated version.
>
> > While I agree that the proposed method visually seems to yield better calibrated uncertainty estimates than MM in Fig. 8, I would also like to see a quantitative evaluation here.
>
> Following the procedure used by Detlefsen et al. (2019) to generate Figure 3 in their paper, we have now added a quantitative evaluation of uncertainty estimates (see Figure 8 (f)). This plot further supports that our method yields well-calibrated uncertainty estimates in comparison to moment matching.
>
> > The tasks are low-dimensional (1D, 3D). I would be interested in results on more complex, higher dimensional problems.
>
> We are in the process of training variational auto-encoders on MNIST, similar to Detlefsen et al. (2019). We will update with another response once those results are ready. This would correspond to 784 dimensions. Do you have any other problems in mind for this?
>
> > Shouldn't Eq. 6 contain a factor of $Z$?
>
> You are correct, thank you for catching this. We modified the equation and statement to read “the gradient of the NLL is proportional to the gradient of the MSE loss under the modified data distribution”.
>
> We also thank you for spotting the other minor issues you reported!

---

> ### Author Response · Authors · 2021-11-22
> **High-Dimensional Regression Experiments**
>
> Dear reviewer gjks,
>
> we now uploaded a revision with experiments with variational autoencoders on MNIST and FashionMNIST. In addition, we evaluate on depth map regression from images on the NYUv2 dataset. See the general answer for a discussion.

---

### Official Review · Reviewer_5Jtp · 2021-11-02

**Correctness:** 3
**Technical Novelty And Significance:** 3
**Empirical Novelty And Significance:** 2
**Recommendation:** 6
**Confidence:** 5

**Main Review:**

### Strengths:
- The NLL loss scaling "the gradient of badly-predicted points, effectively under sampling those points" is an interesting perspective I have not yet seen. The conventional focus, based on my reading of the literature, is the infinite gradient problem (a perfect mean estimate is impossible to obtain using NLL because variance tends towards 0).

### Weakness:
- Figure 1 attempts to illustrate the problem of fitting a neural network to a noisy sinusoid using a NLL loss. However, a few things could be improved. Can the author(s) please confirm if the left panel is single run? If so, does rerunning the code under different initialization move the region with poor predictive mean estimates around? Lastly, it probably would be more clear to show the MSE predictive means in addition to its low RMSE (yes, RMSE is low enough to suggest a perfect fit, but it reduces a reader's burden).
- Claiming the identification of "high dependence of the gradients on the predictive variance as the primary culprit" for NLL failures as a contribution is quite bold. Takahashi, et. al (2018) and Stirn, et al (2020) both discuss this issue, but admittedly with a different perspective--neither claims it as a contribution.
- "We observe a drastic difference between the NLL loss and the MSE/moment matching loss." I understand that the the MM loss for the mean is the same as NLL, but the MM loss also has a variance loss term. How then is 6.b both MSE and MM?
- The second paragraph of section 4.2 mentions not yet seen experimental results. I recommend moving discussion of results to appear after the experiments.
- The author(s) seemingly contradict themselves several times as to whether MSE is a fundamentally different loss from their $\beta = 1$.
	1) "For β = 1 the gradient w.r.t. μ in Eq. 10 is equivalent to the one of MSE. However, for the variance, the gradient in Eq. 11 is a new quantity with 2σ2 in the denominator."
	2) "As expected, the same holds for the mean squared error loss (MSE) which is recovered for β = 1."
	3) The results (figure 8 and table 1) cast  ($\beta = 1$) as MSE, but their loss term has an added variance term.
- The author(s) should have MSE in their loss table since I don't believe ($\beta = 1$) is equivalent. Please correct me if I am missing something!
- The author(s) do not compare to ANY of the cited works, which often outperform them on the UCI regression tasks.

**Summary Of The Paper:**

The author(s) attribute the under-fitting of predictive means by neural networks that parameterize heteroscedastic Gaussian likelihoods to 1) initial inability to break symmetry and 2) Gaussian negative log likelihoods tendency to down weight poorly predicted points. The authors note the latter effectively under samples those points with poor predictive means, thus inhibiting their fit in a rich-get-richer scheme. The author(s) propose two alternative loss functions: moment matching (MM) and $\beta$-exponential variance estimate with $\beta \in [0,1]$. The author(s) demonstrate their $\beta$-exponential generally outperforms NLL ($\beta = 0$) and MSE.

**Summary Of The Review:**

I think this paper offers a fresh perspective on NLL loss failures as I mentioned in my main review. However, their only support for this a contrived toy example over a single run. The paper would be much stronger if:
- The author(s) could apply their analysis to identify either of the two conditions from section 3 in a real world data set.
- Their proposed methods offered comparable (using simpler methods) or superior performance to related works.
Unfortunately, without either of these, the paper's potential significance remains limited.

---

> ### Author Response · Authors · 2021-11-18
> **Response to Review (1/2)**
>
> Thank you for your constructive and detailed review. Below we address your questions / concerns. We start with your major concerns first.
>
> > The paper would be much stronger if: (1/2) The author(s) could apply their analysis to identify either of the two conditions from section 3 in a real world data set.
>
> We agree. In Fig. S4 of the revised version, we now provide evidence for the effective undersampling issue that occurs due to variance-weighting when optimizing the NLL loss (discussed in Sec. 3.2) on the Fetch-PickAndPlace task. This new figure shows how the distribution of effective sampling probability evolves during training (over a fixed set of training points) and compares that against a proxy "oracle": the distribution of effective sampling probabilities when using the squared residuals from a model trained with the MSE loss. The mismatch between the two distributions demonstrates that optimizing the NLL loss drastically undersamples in comparison to the reference, effectively never sampling some data points.
>
> > The paper would be much stronger if: (2/2) Their proposed methods offered comparable (using simpler methods) or superior performance to related works.
>
> We have now added new results using baselines from two prior works (Detlefsen et al., 2019; Stirn & Knowles, 2020). We also added three more datasets of the UCI benchmark (carbon, superconductivity, wine-white). For UCI, see Table 1 in the revised version for a summary of results, and tables S1 and S2 for full results w.r.t. predictive log-likelihood and RMSE. For the dynamics experiments, see Table 2.
>
> We find that baselines based on the Student-t distribution (xVAMP, VBEM) tend to have better predictive log-likelihood than beta-NLL, although there is often no statistical significance. We conjecture this is because the Student-t distribution might be better suited to model real-world data than the Gaussian, especially in low-data regimes. For RMSE, beta-NLL is mostly on-par or better than the baselines. At the same time, our method is very simple to implement and computationally lightweight compared to xVAMP and VBEM, which require the costly evaluation of the prior and Monte-Carlo sampling at each training step.
>
> > Can the author(s) please confirm if the left panel [of Figure 1] is a single run? If so, does rerunning the code under different initialization move the region with poor predictive mean estimates around?
>
> Yes, Figure 1 (left) shows a single run. However, running the code under different initializations leads to qualitatively the same result. To illustrate this, we now show 3 additional trials in Figure S1. We also changed the right panel of Fig. 1 to show mean and standard deviation over 10 independent trials instead of individual curves.
>
> > Lastly, it probably would be more clear to show the MSE predictive means [in Figure 1] in addition to its low RMSE (yes, RMSE is low enough to suggest a perfect fit, but it reduces a reader's burden).
>
> To demonstrate that MSE loss indeed learns an (almost) perfect fit, we now added a dashed line to the right panel of Fig. 1 indicating the RMSE of an optimal fit.
>
> > Claiming the identification of "high dependence of the gradients on the predictive variance as the primary culprit" for NLL failures as a contribution is quite bold. Takahashi et al. (2018) and Stirn et al. (2020) both discuss this issue, but admittedly with a different perspective--neither claims it as a contribution.
>
> We have now modified the statement of our contribution to be more specific, reflecting that the dependence of gradients on predictive variance has been generally known as a cause of instability in optimizing the NLL loss (citing the two references you pointed out) and that our paper provides *a fresh perspective on how this dependence can further be problematic*. (See paragraph 1 of **Summary of contributions** on page 2.)

---

> > ### Author Response · Authors · 2021-11-18
> > **Response to Review (2/2)**
> >
> > > "We observe a drastic difference between the NLL loss and the MSE/moment matching loss." I understand that the MM loss for the mean is the same as NLL *(we think you meant MSE)*, but the MM loss also has a variance loss term. How then is 6.b both MSE and MM?
> >
> > Figure 6b shows MSE for estimating the mean only (i.e. solely optimizing $L^\mu_{\mathrm{MM}}$; moment matching only for the mean and without variance learning). We have now clarified this in the text.
> >
> >
> > > The author(s) seemingly contradict themselves several times as to whether MSE is a fundamentally different loss from their $\beta = 1$.
> > > The author(s) should have MSE in their loss table since I don't believe ($\beta = 1$) is equivalent. Please correct me if I am missing something!
> >
> > We apologise for the confusing implications of equivalence between MSE and $\beta$-NLL with $\beta = 1$ at times. What is in common between the two variants is *their gradient w.r.t. the predictive mean*. When discussing MSE we only train a mean estimator, while when discussing $\beta$-NLL we always jointly train mean and variance estimators (single network with two heads).
> > We have now modified the text and figures to fix this issue; i.e. assertions of equivalence between $\beta=1$ and MSE are now removed and discussions of similarity are now made more specific. We also now ran the UCI datasets using the MSE loss and added the results to the corresponding tables.
> >
> >
> > > The author(s) do not compare to ANY of the cited works, which often outperform them on the UCI regression tasks.
> >
> > We have now added new results using baselines from two prior works (Detlefsen et al., 2019; Stirn & Knowles, 2020) (see answer in response 1/2 above), evaluated in a unified way. For UCI regression in particular, we would like to note that our results are not directly comparable to the reported numbers in previous work, as 1) we use different splits in cross-validation, and results on UCI are known to be sensitive to the exact splits used, and 2) both Detlefsen et al. (2019) and Stirn & Knowles (2020) report numbers resulting from early stopping on the validation set, whereas we report numbers from a separate test set, which negatively affects our results in comparison.

---

> > > ### Comment · Reviewer_5Jtp · 2021-11-30
> > > **Thank you**
> > >
> > > Thank you for addressing my concerns. I will update my score accordingly.

---

### Official Review · Reviewer_5nZv · 2021-11-02

**Correctness:** 3
**Technical Novelty And Significance:** 3
**Empirical Novelty And Significance:** 3
**Recommendation:** 6
**Confidence:** 4

**Main Review:**

**Strong Points**

- The discussion of the optimization issue caused by the NLL is very clear with nice synthetic examples.
- The paper is well written and mostly easy to follow
- Table 2 shows clear improvements in performance over the standard NLL approach


**Weak Points**

- Section 3.1 is difficult to follow. Figure 4 suggests the measurement in Equation (5) is important, but there is a lack of intuition for its significance.
- No ablation study comparing other approaches to solving this problem.


**Questions**

- I wonder how optimizer dependent this issue is. Which optimizers were used and how were they tuned?
- There is a long line of work about “critically” initializing neural networks so that their curvature grows exponentially with depth. I wonder if this is related to Section 3.1. The authors might consider looking at Poole et al., “Exponential expressivity in deep neural networks through transient chaos” and Raghu et al., “On the expressive power of deep neural networks.”
- What is the noise distribution assumed by the beta-NLL? I understand gradients are not taken with respect to the beta-NLL, but I’m curious whether part of the issue is that the Gaussian assumption on the noise is inappropriate.
- How is the bolding chosen in Table 1. It seems like there’s significant overlap between the performance of the different methods.


**Additional Feedback**

- The notation in Equation (9) is confusing. The expectation is over the random variable X, but X also appears outside of any expectation in the stop-gradient operation.


**Summary Of The Paper:**

**Summary**

For regression tasks with heteroscedastic noise, a Gaussian model whose mean and variance are a function of the input is frequently used. Neural networks are then used as function approximators for the mean and variance and are fit using the negative log-likilhood. The paper identifies a problem with this approach: optimization can get stuck in configurations where the predicted mean is far from the true mean, as this is compensated for by a high predicted variance that also stalls learning. By adjusting the NLL objective, the authors compensate for the problem.

**Summary Of The Review:**

**Recommendation**

I like aspects of this paper, but there are some major outstanding questions. Currently, it does not meet the bar for acceptance.

---

> ### Author Response · Authors · 2021-11-18
> **Response to Review**
>
> Thank you for your constructive and detailed review. Below we address your questions / concerns.
>
> > Section 3.1 is difficult to follow. Figure 4 suggests the measurement in Equation (5) is important, but there is a lack of intuition for its significance.
>
> The Jacobian variance in equation 5 is an approximate measure of local curvature of the neural network’s feature space. In Figure 4, we use this measure to show why the neural network resorts to a linear function on some parts of the input space (compare also with Figure 3), which triggers the increase of predictive variance and the self-reinforcing behavior described in Sec. 3.2. We would like to make an effort to make Section 3.1. more clear -- could you point out which parts are unclear?
>
> > No ablation study comparing other approaches to solving this problem.
>
> We have now added new results using baselines from two prior works (Detlefsen et al., 2019; Stirn & Knowles, 2020). We also added three more datasets of the UCI benchmark (carbon, superconductivity, wine-white). For UCI, see Table 1 in the revised version for a summary of results, and tables S1 and S2 for full results w.r.t. predictive log-likelihood and RMSE. For the dynamics experiments, see Table 2.
>
> We find that baselines based on the Student-t distribution (xVAMP, VBEM) tend to have better predictive log-likelihood than beta-NLL, although there is often no statistical significance. We conjecture this is because the Student-t distribution might be better suited to model real-world data than the Gaussian, especially in low-data regimes. For RMSE, beta-NLL is mostly on-par or better than the baselines. At the same time, our method is very simple to implement and computationally lightweight compared to xVAMP and VBEM, which require the costly evaluation of the prior and Monte-Carlo sampling at each training step.
>
> > I wonder how optimizer dependent this issue is. Which optimizers were used and how were they tuned?
>
> All results in the paper are based on the Adam optimizer, with standard hyperparameters settings ($\beta_1=0.9, \beta_2=0.999$). That is, we did not tune the hyperparameters of the optimizers except for learning rates (using grid search). To evaluate the impact of the optimizer, we have tested three canonical optimizers now -- namely Adam, RMSProp, and SGD (with and without momentum) -- on the sinusoidal toy regression problem of Fig. 1. As shown in Fig. S2 in the revised version, the pitfall still persists regardless of the optimizer choice.
>
> > There is a long line of work about “critically” initializing neural networks so that their curvature grows exponentially with depth. I wonder if this is related to Section 3.1. The authors might consider looking at Poole et al., “Exponential expressivity in deep neural networks through transient chaos” and Raghu et al., “On the expressive power of deep neural networks.”
>
> Thank you for mentioning this work. We think that this is related to Sec. 3.1 in the sense that an initially non-linear (i.e. curved) feature space prevents the optimization with NLL from “getting stuck” in some place, stopping the self-reinforcing behavior described in Sec. 3.2. Relating to depth, we had the experience that the issue could be somewhat alleviated with larger depth (cmp. Fig. 7). Studying what constitutes suitable initializations and how they impact NLL training is an interesting direction for future work. Nevertheless, we think that having a robust way to train NLL without the need for careful initialization is an important addition to the ML toolset.
>
> > What is the noise distribution assumed by the beta-NLL? I understand gradients are not taken with respect to the beta-NLL, but I’m curious whether part of the issue is that the Gaussian assumption on the noise is inappropriate.
>
> The noise distribution for $\beta$-NLL is also Gaussian, just as in the standard Gaussian NLL. The modification of $\beta$-NLL only alters the training dynamics and not the modeled noise distribution.
>
> > How is the bolding chosen in Table 1. It seems like there’s significant overlap between the performance of the different methods.
>
> We initially bolded all methods within one standard deviation of the best method, but agree this was not very informative due to the overlap. In the revised version, Table 1 now reports the number of *statistical ties* across 12 UCI datasets, where *ties* gives the number of datasets for which the method can not be statistically distinguished from the best method. We used a two-sided Kolmogorov-Smirnov test for this purpose.
>
> > The notation in Equation (9) is confusing. The expectation is over the random variable X, but X also appears outside of any expectation in the stop-gradient operation.
>
> Thank you for pointing this out. We agree, the notation in Equation (9) was incorrect. We corrected it such that X only appears inside the expectation.

---

> > ### Comment · Reviewer_5nZv · 2021-11-26
> > **Increased score**
> >
> > I am grateful to the authors for all their additional work. They have addressed the questions I raised during my review, so I will raise my score. The reason I am not increasing it further is that the results based on the Student-t distribution complicate the story. It seems like some of the deficiency of the standard Gaussian-noise NLL approach are due to model misspecification rather than problems with optimization.

---

> > > ### Author Response · Authors · 2021-11-27
> > > **Response to the Reviewer**
> > >
> > > We are happy that you are satisfied with our additional investigations.
> > >
> > > While we agree that the Student-t results give hints to study different noise models, this is not the core point of the paper:
> > >  1.  We showed that the issue happens in our toy task where assuming Gaussian noise was the ideal choice (as the task by design featured Gaussian noise).
> > >  2. If the underlying noise distribution is not well-described by Gaussian noise, still we should aim to get as good of a Gaussian fit as possible (measured by the NLL).
> > >  3. The Gaussian noise case is defacto standard in many applications and methods (such as VAEs, differentiable Kalman filters, stochastic policies, etc) and thus we believe our contribution is very valuable to the community.
> > >
> > > We would like to understand whether this addresses your concern or whether there are other aspects we missed.

---

### Author Response · Authors · 2021-11-18
**General Response to Reviewers**

We thank the reviewers for their time and thorough feedback. We uploaded a first revision of our paper taking into account many of your suggestions. Here we briefly outline the newly added results:

- Several baselines (Detlefsen et al., 2019; Stirn & Knowles, 2020) on UCI datasets and the dynamics datasets (Tables 1, 2, S1 and S2).
- Three new UCI datasets (carbon, superconductivity, wine-white) (Tables S1 and S2).
- Evaluation of calibration of predictive variance for the heteroscedastic sine dataset (Figure 8 (f)).
- Several more repetitions of the experiment of Figure 1 (Figure S1).
- Analysis of undersampling behavior on FetchPickAndPlace (Figure S4).

We also would like to report the following errors that we discovered and corrected in this version:

- The data loading code for the sensitivity analysis (Figure 9) was erroneous in not using the final training results. The new version of Figure 9 shows the correct result, which is, however, qualitatively the same. The conclusions we drew from the analysis are not affected by this.
- In the UCI experiments, we trained only on 1D targets instead of on 2D for the datasets: energy and naval. This was due to a bug in the dataset configuration. We retrained all methods on the correct dataset and now report the updated results.

### References:
- Detlefsen et al., 2019: Reliable training and estimation of variance networks. NeurIPS 2019.
- Stirn & Knowles, 2020: Variational Variance: Simple, Reliable, Calibrated Heteroscedastic Noise Variance Parameterization. arXiv:2006.04910.

---

### Author Response · Authors · 2021-11-22
**Another Revision with Image-based Experiments**

We would like to thank reviewer 5Jtp for suggesting to conduct experiments with high-dimensional data. We have now performed several more experiments:
- Variational Autoencoders with probabilistic decoders (see Sec. A.4, Tab. S3 and Fig. S5) on
  - MNIST
  - Fashion MNIST
- Depth-map prediction from images on the NYUv2 dataset (see Sec. A.5, Tab. S4, Fig. S6)

The training with our $\beta$-NLL loss function works without surprises and the results are competitive with the baselines. Our findings are consistent with our observations on low-dimensional datasets. $\beta$-NLL (with $\beta>0$) reaches lower MSE than normal NLL, while achieving good trade-offs for predictive log-likelihood. Figures S5 and S6 show that predictions made by our loss function are visibly improved compared to the NLL loss, while producing meaningful uncertainty maps.

We provide a PDF highlighting all our changes to the submission version in a convenient way at https://ufile.io/kl057blc.
To summarize, we provide more supporting analysis and a broadly expanded evaluation that shows that our proposed beta-NLL loss works competitively in a variety of diverse settings.

---

### Decision · Program_Chairs · 2022-01-20

**Decision:**

Accept (Poster)

**Comment:**

The paper examines the approach of modeling aleatoric uncertainty by fitting a neural network, that estimates mean and variances of a heteorscedasitic Gaussian distribution, based on log likelihood maximization. The authors identify the problem that gradient based training on the netgative log likelihood (NLL) may result in suboptimal solutions where a high predicted variance compensates for the predicted mean being far from the true mean. To solve this problem, the authors suggest to adjust the log likelihood objective by weighting the log likelihood of each single data point by the corresponding beta-exponentiated variance estimate. This adjusted objective is referred to as beta-NLL.

All reviewers agreed that the identified problem and the proposed solution are interesting, that the paper is well written and organized, and that the contributions are significant and somewhat new. The main criticism was on the side of the empirical evaluation. It was criticized that the empirical analysis did not compare the proposed method to other approaches to solving the same problem, that the identified problem and the proposed method should be also analyzed on high-dimensional data, that the results on the synthetic experiments could be improved by investigating more than a single run and by incorporating the the MSE in corresponding Figure 1,  and that standard errors were not reported.

Based on the reviews the authors added several new experiments and investigations in the revised version of their manuscript to improve their empirical analysis: 1) new experiments on high-dimensional data sets were conducted applying variational autoencoders on MNIST and Fashion MNIST and performing Depth-map prediction from images on the NYUv2 dataset. 2) For comparison several baseline methods were added to the experiments on the UCI and the dynamics datasets. 3) Three more UCI datasets (carbon, superconductivity, wine-white) were included in the empirical analysis. 4) An evaluation of calibration of predictive variance for the heteroscedastic sine dataset was added. 5) Several more repetitions of the experiment represented in Figure 1 were conducted. (6) An analysis of undersampling behavior on FetchPickAndPlace was added. Moreover, the authors reported two errors in their previous experiments that they discovered and corrected.

All reviewers were satisfied with the changes in the revised version and the answers to their specific questions and increased their scores accordingly, now commonly voting for acceptance. The paper should therefore be accepted.